Methods

# Identifying space-resolved proteins of the murine thymus, by combining MALDI-MSI and proteomics

Jennifer T Aguilan[1,2], Carlos Madrid-Aliste[2,3], Joshua Fischer[4], Maria K Lagou[5], Simone Sidoli[2,6] (ID), George S Karagiannis[5,7,8,9,10,11] (ID)

Identifying spatially resolved proteomes has advanced markedly, yet integrating definitive protein identification with precise spatial localization in a single workflow remains challenging. Matrix-assisted laser desorption/ionization mass spectrometry imaging (MALDI-MSI) enables antibody-free mapping of proteins in tissue sections, but its capacity for unambiguous identification is limited. Here, we present a combined MALDI-MSI and liquid chromatography-tandem mass spectrometry (LC-MS/MS) approach to map protein localization and track spatial changes in murine thymus during chemotherapy-induced involution and regeneration. Our workflow incorporates a scoring algorithm (pepBridge) that aligns MALDI-MSI molecular signals with LC-MS/MS identifications, enabling confident assignment of proteins, critical to thymic function. Using this pipeline, we reveal spatiotemporal changes in proteins involved in cell migration, cytoskeletal remodeling, and thymic regeneration. Notably, we identify distinct spatial shifts in nucleoprotein TPR and tubulin-associated chaperone A (TBCA), corresponding to chemotherapy-driven architectural remodeling. Translationally, these findings highlight pathways and targets to promote immune recovery in pediatric cancer patients undergoing cytoreductive therapy. Analytically, this framework advances spatial proteomics by enabling high-confidence protein identification in lymphoid tissues, broadening the potential of translational proteomic research.

# Introduction

The identification of spatially resolved proteomes has been a challenge that only recently met some breakthroughs (Mund et al,

2022). Whereas advanced instrumentation and time-consuming protocols can identify proteomes to resolve single-cell identity and heterogeneity, mass spectrometry imaging (MSI) is still met with limitations in combining protein identification with respective localization in a single experimental run. Matrix-assisted laser desorption ionization-MSI (MALDI-MSI) is a technique that circumvents such issues. MALDI-MSI can perform spatial mapping of peptides or proteins directly on a flash frozen or formalin-fixed paraffin-embedded (FFPE) tissue section. One main advantage of MALDI-MSI over traditional immunohistochemistry is that it does not require antigen-specific antibodies to map locations of peptides or proteins on the tissue. Therefore, it is a very useful molecular histology tool for both screening and multiple detection of new disease biomarkers. However, MALDI-MSI is generally limited because of the low signal-to-noise ratio (Gonzalez-Fernandez et al, 2023) and the limited capability of performing MS/MS fragmentation of candidates, which reduces the specificity of detection (Buchberger et al, 2018). Other limitations of MALDI-MSI include the absence of chromatographic separation, which often results in a bias toward detecting the most abundant proteins, and the use of linear TOF mass analyzers for intact protein imaging, which can lead to detector saturation and a limited effective mass range typically up to ~30 kD, with a theoretical maximum near 100 kD.

In this work, we used MALDI-MSI as an unbiased approach for resolving organ/tissue compartmentalization by studying differential expression of proteins through on-tissue peptide spatial mapping. To serve these needs, we adopted an established mouse model of acute thymic involution (Hun et al, 2020), an organ pathology for which MALDI-MSI has not been previously used to extrapolate biological insights. We focused on the thymus, a central lymphoid organ critical for T-cell development. Whereas MALDI-MSI has been applied to thymus tissue previously for metabolite mapping (Tsuji et al, 2021), lipid imaging (Denti et al, 2021), and calcitriol mapping (Meyer et al, 2024), it has not yet been

[1]Department of Pathology, Albert Einstein College of Medicine, Bronx, NY, USA    [2]Laboratory for Macromolecular and Proteomics Facility, Albert Einstein College of Medicine, Bronx, NY, USA    [3]Department of Systems and Computational Biology, Albert Einstein College of Medicine, Bronx, NY, USA    [4]Bruker Daltonics Corporation, New York, NY, USA    [5]Department of Microbiology and Immunology, Albert Einstein College of Medicine, Bronx, NY, USA    [6]Department of Biochemistry, Albert Einstein College of Medicine, Bronx, NY, USA    [7]Gruss-Lipper Biophotonics Center, Albert Einstein College of Medicine, Bronx, NY, USA    [8]Tumor Microenvironment Program, Montefiore-Einstein Comprehensive Cancer Center, Bronx, NY, USA    [9]Integrated Imaging Program for Cancer Research, Montefiore-Einstein Comprehensive Cancer Center, Bronx, NY, USA    [10]Cancer Dormancy Institute, Montefiore-Einstein Comprehensive Cancer Center, Bronx, NY, USA    [11]Marilyn and Stanely M. Katz Institute for Immunotherapy for Cancer and Inflammatory Disorders, Montefiore-Einstein Comprehensive Cancer Center, Bronx, NY, USA

Correspondence: simone.sidoli@einsteinmed.edu; georgios.karagiannis@einsteinmed.edu

used to map the proteome of the thymus, to our knowledge, especially in a regenerative context. Thus, our study provides a first look at thymic protein distributions via MALDI-MSI, combined with LC-MS/MS identification.

Combining MALDI-MSI with LC-MS/MS-based identification is an approach with significant precedent in the literature. For instance, Lemaire et al (2007) pioneered on-tissue trypsin digestion followed by LC-MS/MS of tissue extracts to identify proteins observed by MALDI-MSI. Later, Fata et al (2015), Quanico et al (2017), and Quanico et al (2017) presented variations of the protocol. In 2020, Dewez et al (2020) introduced laser capture microdissection guided by MSI in the procedure. More recent efforts (Guo et al, 2024) have introduced software for automating such integrations with false discovery controls. Our work builds upon this foundation: we apply the approach to mapping the murine thymus proteome, an organ not previously investigated by MALDI-MSI in a proteomic context, and we introduce a scoring-based informatics pipeline (pepBridge) to facilitate peptide matching on a lower resolution MALDI-TOF platform. The novelty of our study, therefore, lies in the specific biological insights into thymus regeneration and the tailored computational method.

The thymus is a primary lymphoid organ of T-cell development and functional maturation and sits encapsulated above the heart in the mediastinum. The thymus consists of two lobes that divide into lobules, which in turn, are subdivided into two distinct zones, an outer cortex and an inner medulla. These compartments embody functional "niches" that provide essential spatiotemporal signals that regulate key processes of T-cell development, in particular thymocyte homing, survival, and expansion, as well as T-cell receptor chain rearrangement, and the establishment of central tolerance. In both embryonic and non-embryonic thymi, these processes are spatially orchestrated by an intrathymic stromal network of cortical (cTEC) and medullary (mTEC) thymic epithelial cells, as well as by a supporting network of non-epithelial stromal cells, such as fibroblasts, blood vessel endothelial cells and their associated pericytes, etc. All these stromal cells, either epithelial or non-epithelial, reside in and specialize their functions in distinct cortical and medullary thymic compartments (Kato, 1997; Owen et al, 2000; Chidgey & Boyd, 2001; Munoz et al, 2009; Manley et al, 2011; Anderson & Takahama, 2012; Abramson & Anderson, 2017; Takahama et al, 2017; Alawam et al, 2020; Kadouri et al, 2020; Sun et al, 2021a; Han & Zuniga-Pflucker, 2021; James et al, 2021; Bhalla et al, 2022; Nitta, 2022; Cabric & Brown, 2023; Vodopyanov et al, 2025a; Vodopyanov et al, 2025b).

During preadolescence, the thymus plays an essential role in establishing a comprehensive and enduring T-cell repertoire, which is pivotal for systemic immunity, immune surveillance, and the establishment of central tolerance (Cahill et al, 1997; Baran-Gale et al, 2020; Webb & Haniffa, 2023). Notably, this period marks the zenith of the thymus functional capacity, which subsequently experiences a progressive, age-related decline. By early adulthood, the reliance of systemic immunity on the thymus decreases, shifting towards increased dependence on the expansion of peripheral memory T cells (Ostan et al, 2008; Aw et al, 2009; Palmer, 2013; Palmer et al, 2018; Minato et al, 2020; Rane et al, 2022). More importantly, the thymus is particularly vulnerable to a variety of cytotoxic agents, leading to its rapid (acute) thymic involution (ATI)

(Yang et al, 2009; Swami et al, 2012; Chaudhry et al, 2016; Ansari & Liu, 2017; Kinsella & Dudakov, 2020; Lagou et al, 2022). For instance, treatment of cancer patients with cytoreductive chemotherapy leads to both short- and long-term consequences on systemic immunity, in part because of ATI (Fletcher et al, 2009; Ansari & Liu, 2017; Wertheimer et al, 2018; Kinsella & Dudakov, 2020; Lagou et al, 2022). Given the key role of the thymus in immune development in children, ATI emerges as a pressing concern for pediatric cancer patients undergoing cytoreductive treatments, thus necessitating meticulous consideration to alleviate its effects on systemic immunity. Administration of chemotherapy and radiotherapy exacerbates ATI and compromises the immune system, making pediatric patients vulnerable to infections, and challenging immune system recovery post-therapy (Ansari & Liu, 2017; Lagou et al, 2022). Strategies aimed at fostering endogenous thymic regeneration during or after chemo/radiotherapy hold the promise of significantly improving life expectancy and quality-of-life for pediatric cancer survivors. However, progress in understanding thymic recovery post-cytotoxic treatment has been limited, especially because of the lack of systemic approaches such as single-cell transcriptomics and high-throughput proteomics. Advancing these techniques in this field could significantly enhance our ability to restore immune functions and competence in pediatric cancer patients.

In addition, given the histological perplexities of lymphoid organs such as the thymus, it is imperative to implement spatial resolution in such sophisticated analytical techniques, to be able to gain sufficient and valuable information for complicated immune responses. Given that MALDI-MSI circumvents these caveats by generating large sets of imaging spectra, here, we compiled a library of candidates and then used LC-MS/MS analysis of serial or adjacent thymic tissue sections, to create a list of candidate peptides detectable in the samples. We subsequently engineered a peak-matching pipeline with a customizable scoring system, to match LC-MS/MS with MALDI-MSI data, and prioritize candidates based on identification confidence and quantification. In summary, this proposed methodology applies the techniques of MALDI-MSI and LC-MS/MS in parallel thymic slide sections, to identify potential protein markers of thymic compartmentalization and endogenous thymic regeneration, using a mouse model of chemotherapy-induced involution. Compared with more routine proteomics experiments, our pipeline combines the identification of potential candidate biomarkers with feasibility assays in a single shot, to assess the capability of MALDI-MSI in confidently quantifying these signals.

## Results

### Pipeline overview

A graphical summary of the pipeline followed in this work is presented in Fig 1. Briefly, FFPE tissue sections are prepared for protein digestion using a TM sprayer and analyzed by MALDI-MSI (Fig 1A). In parallel, digested peptides from the thymic tissues are collected, dried, and analyzed by LC-MS/MS (Fig 1B). An integration

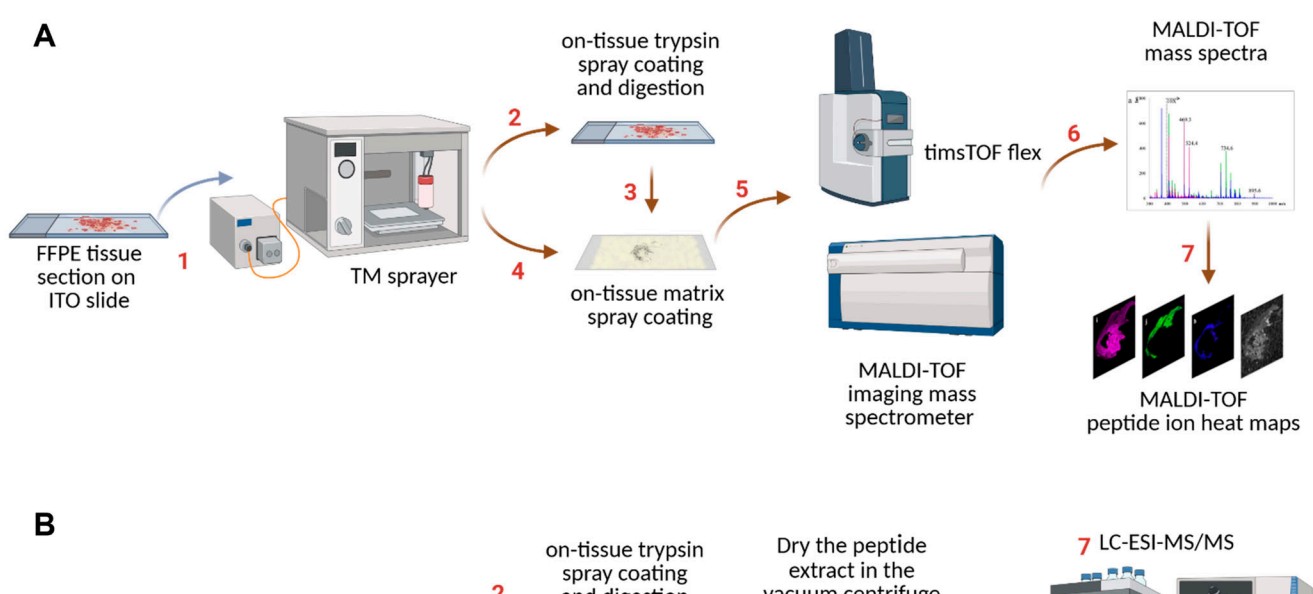

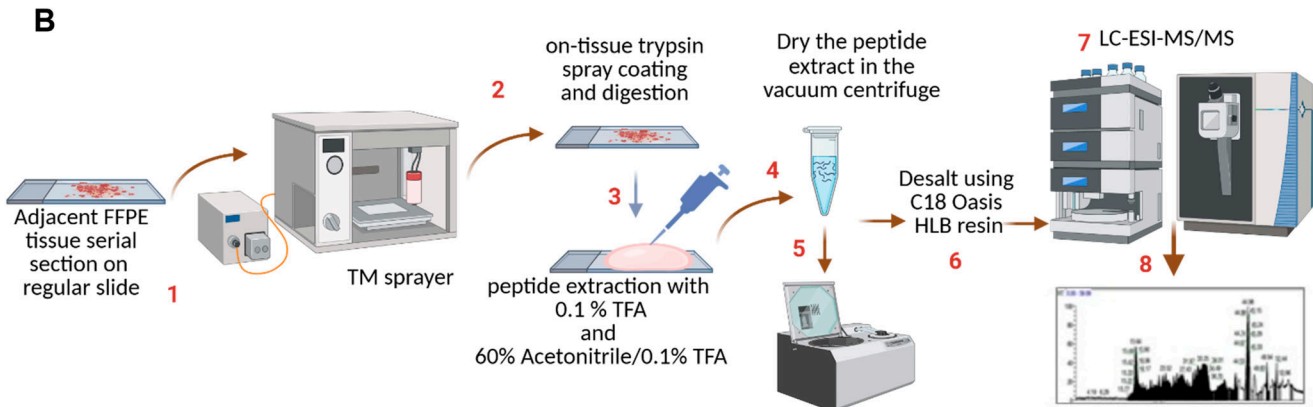

**Figure 1.  Workflow for spatial proteomics analysis of thymic tissues combining MALDI-MSI and LC-MS/MS.**
**(A)** Thymic FFPE tissue sections were prepared for MALDI-MSI analysis, including deparaffinization, enzymatic digestion, and matrix coating. MALDI-MSI was performed to acquire spatially resolved molecular signals across tissue sections, generating ion heat maps for distinct protein distributions. **(B)** Parallel FFPE tissue sections were digested, and peptides were extracted for LC-MS/MS analysis using high-resolution Orbitrap mass spectrometry. Identified peptides were label-free quantified to determine fold-change enrichments. This workflow enabled the identification and spatial mapping of thymic protein biomarkers, facilitating insights into thymic compartmentalization and regeneration.

of these parallel datasets, via our newly developed, customizable algorithm PepBridge, leads to candidate markers of thymic compartmentalization and endogenous regeneration after chemotherapy treatment.

## Mouse model of cyclophosphamide-induced thymic involution

We used digital pathology to examine whole slides of murine thymi (Fig 2A). Under normal conditions, the murine thymus is surrounded by a connective tissue capsule (Fig 2A–C), which is composed of fibroblasts, blood vessels, and collagen fibers (Suster & Rosai, 1990; Moll, 1997; Pearse, 2006; Perez & Moran, 2022). The capsule invaginates to the organ's interior, forming connective tissue trabeculae, also known as "septae" (Fig 2A–D), which correspond to the routes of inbound and outbound blood vessels and nerves (Suster & Rosai, 1990; Moll, 1997; Pearse, 2006; Perez & Moran, 2022). These septae segregate the thymus into basic structural/functional units, thymic lobules (Fig 2B) (Suster & Rosai, 1990; Moll, 1997; Pearse, 2006; Perez & Moran, 2022). There are two distinct zones identified within the thymic lobules, an "outer"

darker zone, which is characterized by densely packed T lymphocytes and is called "cortex," and an "inner" lighter zone, which is mainly comprised by sparser T lymphocytes and is called "medulla" (Suster & Rosai, 1990; Moll, 1997; Pearse, 2006; Perez & Moran, 2022) (Fig 2A–D). Thymic compartmentalization via the connective tissue septae is not complete, as the septae never physically interact with one other in the medullary regions. As such, the medulla is often visualized as a single and relatively continuous entity (Fig 2A), depending on the orientation of the tissue section (Suster & Rosai, 1990; Moll, 1997; Pearse, 2006; Perez & Moran, 2022). A microanatomical region of great functional importance is the corticomedullary junction (CMJ), represented as a relatively distinct border between cortex and medulla (Fig 2E), in which trabecular vessels branch out, creating the CMJ vessels (Fig 2E) (Suster & Rosai, 1990; Moll, 1997; Pearse, 2006; Perez & Moran, 2022). These vessels are opted for the immigration of early thymocyte progenitors (ETPs) in the thymus and for the egress of mature CD4[+] or CD8[+] T cells in the peripheral blood (Suster & Rosai, 1990; Moll, 1997; Pearse, 2006; Perez & Moran, 2022).

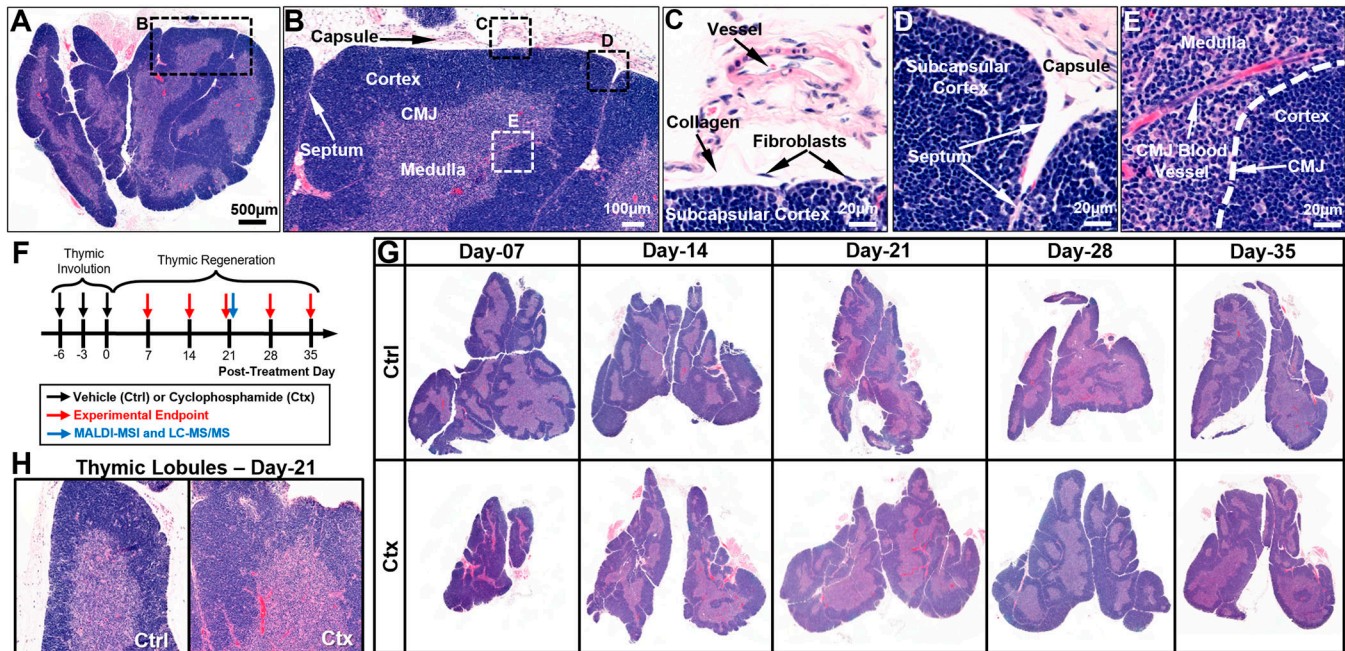

**Figure 2. Mouse model of cyclophosphamide-induced thymic involution.**
**(A, B, C, D, E)** The key histological features and compartments of the murine thymus, using Hematoxylin & Eosin (H&E) staining. **(A)** Whole murine thymus, segmented into lobes, lobules through connective tissue capsule and septa, and compartmentalized into outer cortex and inner medulla. **(A, B)** Magnified insert of the black dotted box in (A), revealing a thymic lobule. **(B, C, D)** Magnified inserts of the black dotted boxes in (B), revealing cellular components of the thymic capsule (C) or the interlobular septa (D). **(B, E)** Magnified insert of the white-dotted box in (B), revealing cellular components of the corticomedullary junction. **(F)** Pipeline of the mouse model of cyclophosphamide-induced thymic involution. The regenerative process is monitored on a weekly basis following the last dose of chemotherapy (day 0) for up to 5 wk (day 35). MALDI-MSI coupled to LC-MS/MS was conducted on day 21, i.e., 3 wk following the last dose of chemotherapy. **(F, G)** Representative H&E sections of whole murine thymi at the distinct timepoints of thymic regeneration after cyclophosphamide insult, as indicated in (F). The rebound hyperplasia is notable on days 21 and 28. **(H)** Investigation of thymic lobules during rebound hyperplasia, reveals the maintenance of the overall lobular architecture, albeit with a vast thickening of the lobules, and the disturbed corticomedullary ratio, mostly attributable to increased proliferation of thymocytes in the cortical zone. CMJ, corticomedullary junction; Veh, vehicle; Ctx, cyclophosphamide.

Acute thymic involution in mice occurs after the administration of cytotoxic agents such as chemotherapy, corticosteroids, antibiotics, and other drugs, and physiologically mimics human responses (Chaudhry et al, 2016; Ansari & Liu, 2017; Kinsella & Dudakov, 2020; Lagou et al, 2022). Cyclophosphamide (Ctx), an alkylating agent that is often administered in the course of chemotherapy treatment, is an established immunosuppressive drug with deliberating consequences on the thymus (Kahri et al, 1965; Anton, 1987; Yoon et al, 1997; Basta-Kaim et al, 2001; Fletcher et al, 2009; Erokhina & Avilova, 2019). To monitor global proteomic changes during endogenous thymic regeneration after chemotherapy, here, we developed a mouse model of cyclophosphamide-induced thymic involution by exposing female FVB/NCrl mice to radical doses of cyclophosphamide (i.e., 200 mg/Kg every 3 d), followed by 5 wk of post-chemotherapy monitoring. This experimental pipeline allowed us to capture precise kinetics of endogenous thymic regeneration after the chemotherapeutic insult (Fig 2F), a process often deemed complex because of the plethora of signaling circuitries regulating thymic remodeling (Dudakov et al, 2012; Wertheimer et al, 2018; Zhan et al, 2019). We noted marked thymic involution during the 1st wk after chemotherapeutic insult, characterized by ambiguous corticomedullary junction and hypoplastic cortex (Fig 2G). Endogenous thymic regeneration was relatively fast, with thymic mass reaching an equilibrium by the 2nd wk after the chemotherapeutic insult (Fig 2G).

Interestingly, there was a relative increase in thymic size by the 3rd and 4th wk of endogenous thymic regeneration (Fig 2G), which is consistent with prior observations of "rebound hyperplasia" in human patients receiving chemotherapy (Sun et al, 2016; Chen et al, 2017; Arpaci & Karagun, 2018; Deniz et al, 2020; Lazaro-Garcia et al, 2021; Qiu et al, 2021; Ayyildiz et al, 2022; Franke et al, 2023; Gulati & Giulino-Roth, 2023). Rebound hyperplasia, whose peak was at 3rd wk after chemotherapeutic insult (Fig 2G), was characterized by excessive cortical hyperplasia and a disanalogous medullary compartment, resulting in an evident increase in the corticomedullary ratio (Fig 2H). A "recoil" phase between 4th and 5th wk after the chemotherapeutic insult, during which thymic size reverted to its homeostatic size, was captured as the final step of endogenous thymic regeneration (Fig 2G). Overall, the impact of cyclophosphamide on the murine thymus and the associated kinetics (Fig 2G) are consistent with prior reports (Basta-Kaim et al, 2001; Fletcher et al, 2009; Goldberg et al, 2010; Sun et al, 2015; Hun et al, 2020).

### Hierarchical clustering of MALDI-MSI acquired spectra from the murine thymus

To identify biomarkers predictive of thymic regeneration, we leveraged the detailed kinetics of endogenous thymic regeneration after Ctx-induced injury. We focused on the 21-d post-Ctx

timepoint for several reasons. First, although "rebound hyper-plasia" is commonly observed in patients after chemotherapy or other anticancer treatments, no established biomarkers currently gauge thymic functionality during this condition. Second, to our knowledge, a spatially resolved proteome map has never been generated for the regenerating thymus, particularly during re-bound hyperplasia. With these considerations, we collected thymi from vehicle-treated (Ctrl) and Ctx-treated mice at 21 d after ending their respective treatments (Fig 2F). The tissues were processed as FFPE samples and subjected to MALDI-MSI after our established pipeline (Fig 1A). From this analysis, we obtained comprehensive mass-to-charge (m/z) maps, which were pro-cessed using Fleximaging v4.1, to extract signals uniquely defining specific regions of interest. To correlate these molecular patterns with anatomical features, the same thymic sections used for MALDI-MSI were also stained with hematoxylin and eosin (H&E). This enabled the direct superimposition of molecular images onto histological structures.

We then performed unsupervised hierarchical clustering of the spectra to identify unique features represented by distinct nodes in the resulting dendrogram (Fig S1). This approach generated segmentation maps from the acquired tissue images, assigning specific colors to groups of pixels that shared similar spectral signatures. Representative clustering outputs across thymic re-gions are shown in Fig S1B–F, confirming that spectral grouping reliably reflects anatomical compartmentalization. We then extracted three highly unique m/z values that defined each cluster, specifically those at 946, 1,337 and 1,454 Da, and generated their corresponding extracted ion images. Interestingly, each of these three signals represented a different cluster and compartment of the thymus (Fig S2). Assigning a distinct color to each signal generated individual ion heat maps, which, when overlaid, dem-onstrated that the spatial relationships among the three signals were preserved. Reconstructing molecular images from the 946, 1,337, and 1,454 Da signals revealed compartment-specific ex-pression patterns within the thymus (Fig 3A–D). For example, the 1,454 Da signal delineated the thymic capsule, various stromal elements, and surrounding connective tissues (Fig 3B). The 946 Da signal marked elements expressed predominantly in the medulla, though also present in the cortex (Fig 3C). In contrast, the 1,337 Da signal was expressed throughout the thymus, with a stronger presence in the cortex (Fig 3D).

Interestingly, some molecular images also revealed differences between normal thymus and thymus undergoing rebound hy-perplasia after Ctx insult. The 946 Da signal, for instance, showed a clear corticomedullary distinction in Ctrl but exhibited a less defined pattern in the Ctx-treated thymus (Fig 3C). This suggested that the corresponding protein may be reduced or functionally altered during the endogenous regenerative process. In summary, our MALDI-MSI approach accurately mapped thymic compart-mentalization, specifically demarcating the thymic capsule and distinct corticomedullary zones, whereas also revealing histo-logical changes in thymic architecture under experimental conditions.

Our next step was to identify the major discriminating signals uncovered by these analyses. Given that unsupervised hierarchical clustering of the MALDI-MSI data yielded three major spectral classes, characterized by m/z 946, 1,337, and 1,454 as distinguishing signals for the thymic medulla, cortex, and capsule regions, re-spectively (Fig 3A–D), we cross-referenced the m/z values with our LC-MS/MS identifications, to link these cluster-specific ions to proteins. Notably, each could be assigned to a peptide: m/z 946 matched the [M+H]+ of the Nucleoprotein TPR (sequence ESLLAEQR, monoisotopic 945.5 Da), m/z 1,337 corresponded to a peptide from the Ras-related protein Rab14 (sequence GAA-GALMVYDITR, 1337.7 Da), and m/z 1,454 aligned with Serotransferrin (sequence SKDFQLFSSPLGK, 1453.8 Da). These tentative identifi-cations are reported in Table S2.

## LC-MS/MS proteomics analysis of peptides collected from tissue digestion

One challenge in MALDI-MSI is the assignment of precise identities to the obtained signals. To address this challenge, we performed proteomic analysis of extracted peptides from the tissue slides, which consisted of an online LC-MS/MS using data-dependent acquisition (Fig 1B). The total number of proteins and peptides for the thymus tissue samples are shown in Fig 4A and B. Identified peptides were label-free quantified for each tissue section cor-responding to a time point to determine the fold-change en-richment between the treated (Ctx) and vehicle-treated conditions (Ctrl). With this analysis, we obtained a list of identifiable targets present in thymic tissues, which could be subsequently matched to the MALDI-MSI discriminant peaks. Fig 4C shows representative ion chromatograms and peptide identification metrics demonstrating reproducible LC-MS/MS detection across biological replicates.

## MALDI-LC-MS/MS peak-matching pipeline (pepBridge) combined with a four-metric scoring system

The major challenge in this step is the issue related to the rela-tively low resolution of MALDI-TOF mass spectrometry (40,000 at m/z 400) compared with the higher resolution provided by orbi-trap mass spectrometry (120,000 at m/z 200) used in the LC-MS/MS experiment. With a mass tolerance for MALDI-MSI spectra set at ±1 Da, it frequently happened that more than one peptide candidate identified in the LC-MS/MS runs matched with the potential m/z. For this reason, we developed PepBridge (Fig S3), a MALDI-LC-MS/MS peak-matching pipeline based on four rationally designed metrics based on: (1) delta fold change; (2) delta mass; (3) MALDI intensity and (4) matched image or ion heat map. Each peptide mass identified from LC-MS/MS and matched to MALDI received a score from each metric. The sum of the peptide scores from metrics 1–3 was used to rank the probabilities of peptide match. The highest score corresponded to a top candidate peptide with the highest matching probability, and whose sequence belonged to a particular protein. As a positive control, we used trypsin that was sprayed on the MALDI plate and selected regions of interests at specific locations outside the tissue sections that were expected to contain only trypsin (Figs S1A, A', and A" and S2A and A'). Indeed, the PepBridge scoring system revealed that the tryptic peptides were the most enriched, obtaining the highest score and validating our pipeline (Fig S4 and Table S1).

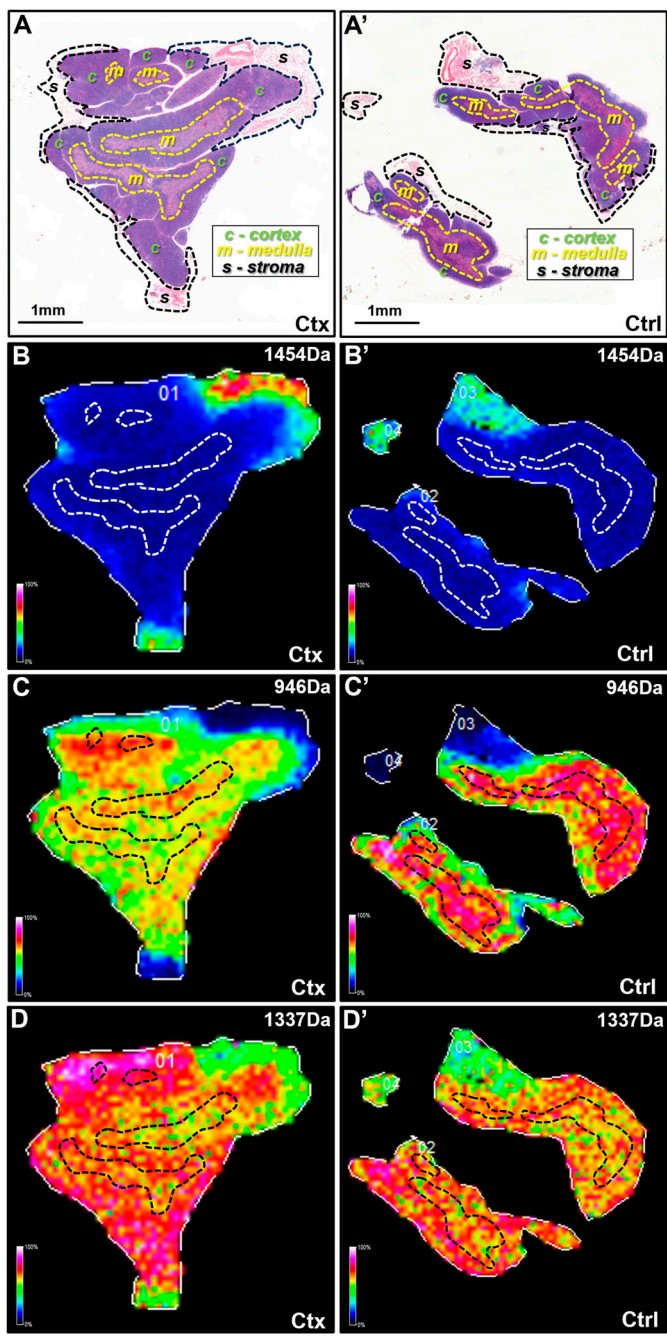

**Figure 3. Assessing protein distribution in thymic compartments.**
**(A)** Transverse sections of murine thymi from animals receiving either cyclophosphamide (Ctx) or vehicle (Ctrl) and then sacrificed 3 wk following termination of treatments, and stained with H&E. Dotted black lines indicate various thymic compartments, such as perithymic stroma (s), cortex (c), and medulla (m), identified by their unique histological features. **(B, B', C, C', D, D')** Molecular images constructed from selected masses show a specific signal at 1,454 Da (B, B'), 946 Da (C, C'), and 1,337 Da (D, D'). These protein ions signals demonstrate unique and diverse compartmentalization within the thymic structures, with $m/z$ 1,454 preferentially localized to the thymic capsule and the perithymic stromal tissues (B), whereas $m/z$ 946 (C) and $m/z$ 1,337 (D) mostly found in the thymic parenchyma with stronger localization to the medullary or cortical compartments, respectively. However, disturbances in the compartmentalization of the $m/z$ 946 and $m/z$ 1,454 signals are captured in the

The imaging data from Veh-treated (Ctrl) and Ctx-treated thymic sections were processed using the PepBridge MALDI-LC-MS/MS peak-matching pipeline. To enable comparison between low- and high-resolution MALDI data, a ±1 Da mass filter was applied. Analysis of the UltrafleXtreme dataset (~15,000 resolving power at $m/z$ 1,000) yielded ~23,000 initial MALDI-LC-MS/MS matches. This high number reflected both the relatively low mass resolution of the MALDI-TOF/TOF platform producing broader, less precise $m/z$ peaks than newer systems such as multi-reflectron TOF instruments (Verenchikov et al, 2024) or the timsTOF flex (~60,000–80,000 resolution at $m/z$ 400), which provides medium resolving power compared with Orbitrap or FT-ICR platforms. Regarding the combinatorial nature of the matching process, each MALDI peak (~300–500 per average spectrum) could fall within the ±1 Da tolerance of many of the ~6,500 LC-MS/MS-identified peptides. To refine the list, we removed matches involving low-intensity MALDI peaks (signal-to-noise < 2), reducing the dataset to 676 matches (Fig 4D; Table S2). These 676 MALDI peaks each had at least one plausible LC-MS/MS match and were subsequently ranked using our four-metric scoring system (Fig 4E–G). The top-ranked candidates were then carried forward for biological in-terpretation and validation. When the same tissue sample was analyzed on the higher resolution timsTOF flex, the initial match count was ~2,500. This was ~10-fold lower than for the Ultra-fleXtreme, reflecting a reduced peak broadness and ambiguity at higher resolution.

To combine disparate metrics on a common scale, each was transformed into a bounded score (Fig 4E–G). **Metric 1** ($\Delta\log_2$ fold change between MALDI-MSI and LC-MS/MS) uses a Gaussian function centered at zero, rewarding small discrepancies with higher scores (0–10). Whereas designed for control-treatment comparisons, it can be adapted to any two groups (e.g., region-to-region or PCA clusters) by calculating a pseudo-fold change. **Metric 2** (mass difference) is scored via a sigmoid (0–100), sharply penalizing offsets >0.1 Da. A ±1 Da tolerance maximized sensitivity but occasionally matched isotopic peaks; such cases were flagged rather than removed to avoid false negatives, especially with lower resolution MALDI data. With higher resolution platforms (e.g., timsTOF flex), Metric 2 readily separates isotopes, with false positives scoring near zero. **Metric 3** (MALDI intensity rank) applies a sigmoid (0–5), giving modest weight to signal strength to avoid over-penalizing low-abundance peptides when favoring robustly detectable features. Together, these non-linear transformations ensure that minimal fold-change or mass errors yield near-maximal scores, whereas larger discrepancies or weak signals sharply reduce confidence. Ultimately, the Peptide Score is the sum of scores from Metrics 1–3 for each peptide (Table S2). This corresponds to the rank of the peptide or its matching probability. Interestingly, we found that ranking the peptides within its protein group proved to be more effective in selecting the top peptide

Ctx-treated thymi, suggesting spatiotemporal changes of the corresponding protein ions. **(A, B, D)** Dotted black boxes in (B) through (D) indicate the medullary compartments, as co-registered from the accompanying H&E in (A).

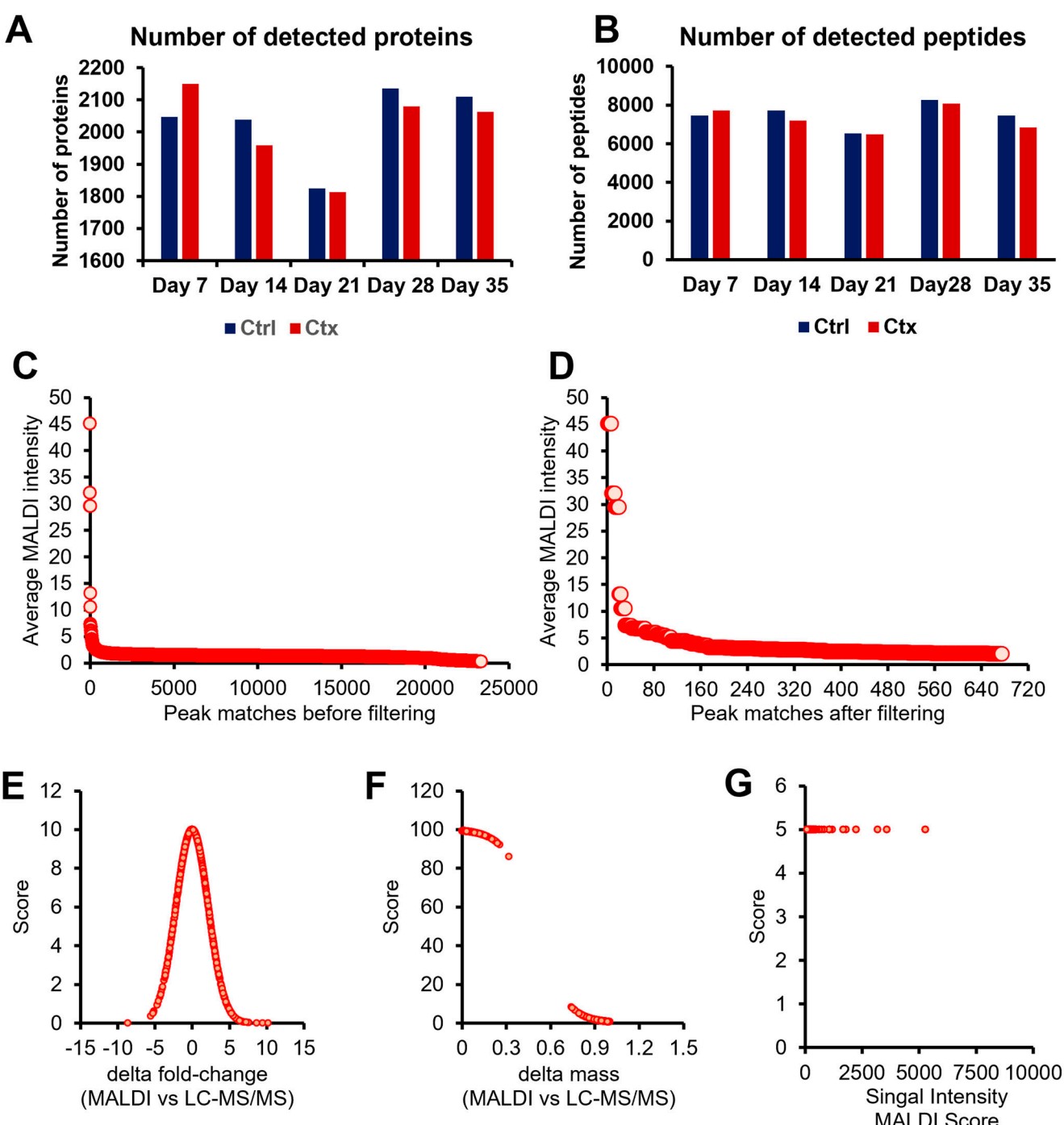

**Figure 4. Proteomic analysis of thymic peptides from cyclophosphamide-treated and control mice.**
**(A, B)** Total number of proteins and (B) peptides identified by LC-MS/MS from thymic tissue sections of cyclophosphamide-treated and vehicle-treated mice. **(C, D)** Number of MALDI peaks sorted by intensity and matched with LC-MS data (~23,000 matches) before and (D) after the filtering with the three scoring criteria of PepBridge (676 peptide matches). **(E, F, G)** Scoring metrics used in the PepBridge pipeline: delta fold change between MALDI and LC-MS/MS (E), delta mass difference (F), and MALDI intensity (G). Peptides were ranked based on the combined scores from these metrics. The pipeline enabled confident identification of peptide candidates, narrowing the list to 68 high-confidence peptides associated with thymic compartmentalization and regeneration.

candidates by manually comparing the peptide scores and the log fold changes within a protein group.

In the updated timsTOF flex workflow, **Metric 4** is automated: the top 10 most abundant peptide ion images showing the largest differences between vehicle-treated (Ctrl) and Ctx-treated samples were selected without manual image review (Fig S5). We retained a ±1.0 Da mass tolerance to maximize sensitivity, though this occasionally resulted in duplicate assignments from isotopic

peaks (e.g., both the monoisotopic and +1 Da isotope for Histone H4, Table S2). Whereas more stringent processing such as deisotoping or narrower mass windows could reduce duplicates, these approaches risk false negatives given the current MALDI-MSI resolution. After these filters were applied, the list of candidates was narrowed down to fewer candidates (Table S3 and Fig S5). Additional examples of top-ranking peptide ion distributions across experimental time points are provided in Fig S6, illustrating the reproducibility of high-abundance peptide features in the timsTOF flex dataset. This, to our knowledge, represents the largest spatiotemporally characterized proteome during endogenous thymic regeneration after chemotherapy insult. Protein candidate(s) of interest can be further validated by another experimental technique, such as immunohistochemistry.

### Protein-protein interaction (PPI) analysis reveals three major functional clusters supporting thymic regeneration after cyclophosphamide insult

The candidate proteins identified by PepBridge were analyzed using the STRING database and grouped by k-means clustering (Fig 5). Most proteins were interconnected, forming three major functional clusters, with one small Fga-F3 cluster pair corresponding to blood coagulation and eight unconnected proteins (Ubxn1, Tbca, Numa1, Kansl3, Cbr1, Vac14, Rab14, Marcksl1) lying outside the main networks (Fig 5). A few proteins, such as vimentin (Vim), ribosomal protein L7a (Rpl7a), and heterogeneous nuclear ribonucleoprotein M (Hnrnpm), displayed clear "hub-like" activity, consistent with a distributed network topology rather than reliance on a single dominant regulator (Fig 5), in line with previous models of endogenous thymic regeneration documenting divergent pathways involved in the process (Chaudhry et al, 2016; Kinsella & Dudakov, 2020; Velardi et al, 2021).

Cluster 1 (yellow nodes) includes keratins (Krt5, Krt7, Krt8), vimentin (Vim), proliferation marker Mki67, actin-related proteins (Acta1, Was), myosin heavy chain (Myh9), and regulators such as Rhog and Tgfbi. These proteins maintain thymic stromal scaffolding, especially in cTECs and mTECs, and mediate thymocyte migration and trafficking during regeneration (Serrador et al, 1999; Kuznetsov et al, 2017; Kadouri et al, 2020; Ahmad Mokhtar et al, 2022; Thompson et al, 2022). Some unconnected proteins, such as Marcksl1, have known cytoskeletal roles and may be functionally related despite lacking high-confidence STRING edges (El Amri et al, 2018).

Cluster 2 (green nodes) centers on ribosomal proteins (Rpl23a, Rps18, Rpl7a, Rpl26, Rpl19, Rps9) and factors such as Pkm, Tkt, Sbds, Srp54b, and Impdh2. These support high translational output and efficient targeting of membrane/secretory proteins, which is essential for TEC and thymocyte proliferation post-injury (Sun et al, 2021b; Marchingo & Cantrell, 2022). Metabolic enzymes such as Pkm and Tkt may reflect shifts in energy demand during tissue repair (Im & Hoopes, 1978; Kim et al, 2023).

Cluster 3 (red nodes) contains RNA-binding proteins (Hnrnpm, Hnrnpab, Hnrnpul1, Elavl1, Rbm15, Puf60) and splicing factors (Srsf2, Srsf5, Thrap3), along with regulators such as Ranbp1, Trim28, and SupT16. These coordinate RNA processing, transport, and alternative splicing, functions critical for reprogramming gene expression during thymic regeneration (Geuens et al, 2016; Le et al, 2020; Qi et al, 2021; Thibault et al, 2021; Zhang et al, 2024). Proteins

such as Elavl1 and Trim28 additionally link RNA metabolism to stress response and chromatin remodeling (Ghosh et al, 2009; Papadaki et al, 2009; Zhou et al, 2012).

Overall, the PPI map depicts structural, translational, and RNA-regulatory pathways acting in parallel, underscoring the multifaceted nature of thymic repair.

### Validation of MALDI-MSI KRT8 mapping by immunofluorescence

To validate the PepBridge-based MALDI-MSI data, we focused on keratin 8 (KRT8), a key epithelial marker expressed in both cTECs and mTECs, and KRT5, which is mTEC-specific. Both are critical for thymic homeostasis and regeneration (Lee et al, 2011; Odaka et al, 2013; Gupta et al, 2016). Serial FFPE sections, 5 $\mu$m apart from those analyzed by MALDI-MSI, were stained for KRT8 and KRT5 to co-register molecular and histological data (Fig 6A and B). Whereas minor morphological differences are expected between adjacent sections, overall compartment topology remained consistent. MALDI-MSI revealed KRT8 signal in both cortical (KRT5⁻) and medullary (KRT5⁺) regions of Veh- and Ctx-treated thymi, with stronger expression in medulla and no detectable signal in the capsule or perithymic stroma (Fig 6A and B). On the other side, the KRT5 signal was found in the medulla, but some cortical signal was also present in form of large islets (Fig S7A). Immunofluorescence confirmed these patterns: capsule regions showed no KRT8 expression (Fig 6C) and adjacent brown adipose tissue was also KRT8⁻ (Fig 6D), whereas KRT5 expression mostly labelled the medullary regions (Fig S7B and C), thus matching the MALDI-MSI data. Importantly, these validations of KRT5/8 expression were made both in an adjacent slide to the MALDI-MSI (through co-registration) (Fig 6), thus providing direct evidence of coordination between MALDI-MSI and IF images, as well as in independent thymus slides (Fig S7B and C), thus showcasing the reproducibility of KRT5/8 expression across murine thymi.

At the corticomedullary junction, both methods showed KRT8 expression in the cortex and medulla, with higher medullary abundance (Fig 6E and F), and MALDI-MSI detected a gradual signal decrease from medulla to cortex. In some junctions (Fig 6G and H), MALDI-MSI showed a more uniform KRT8 signal across the boundary. Immunofluorescence of these regions revealed denser cTEC meshwork adjacent to the medulla, consistent with active epithelial remodeling. These uniform-signal niches occurred in both treatment groups but were more common after Ctx insult, likely reflecting rebound hyperplasia during regeneration (21 d post-treatment; Fig 2F–H). Besides distinguishing among thymic compartments, MALDI-MSI was, therefore, accurate in capturing expected protein expression gradients and shifts between compartments.

It should be noted that the IF validation sites were selected based on MALDI-MSI signal patterns, underscoring the method's sensitivity and specificity in detecting spatial variations in protein expression at corticomedullary boundaries and perithymic zones.

### Validation of additional PepBridge-identified proteins by MALDI-MSI and immunofluorescence across thymus compartments

Validation using KRT5/8 was the most logical option given that they are key markers for cTEC/mTEC in the thymus. To further confirm

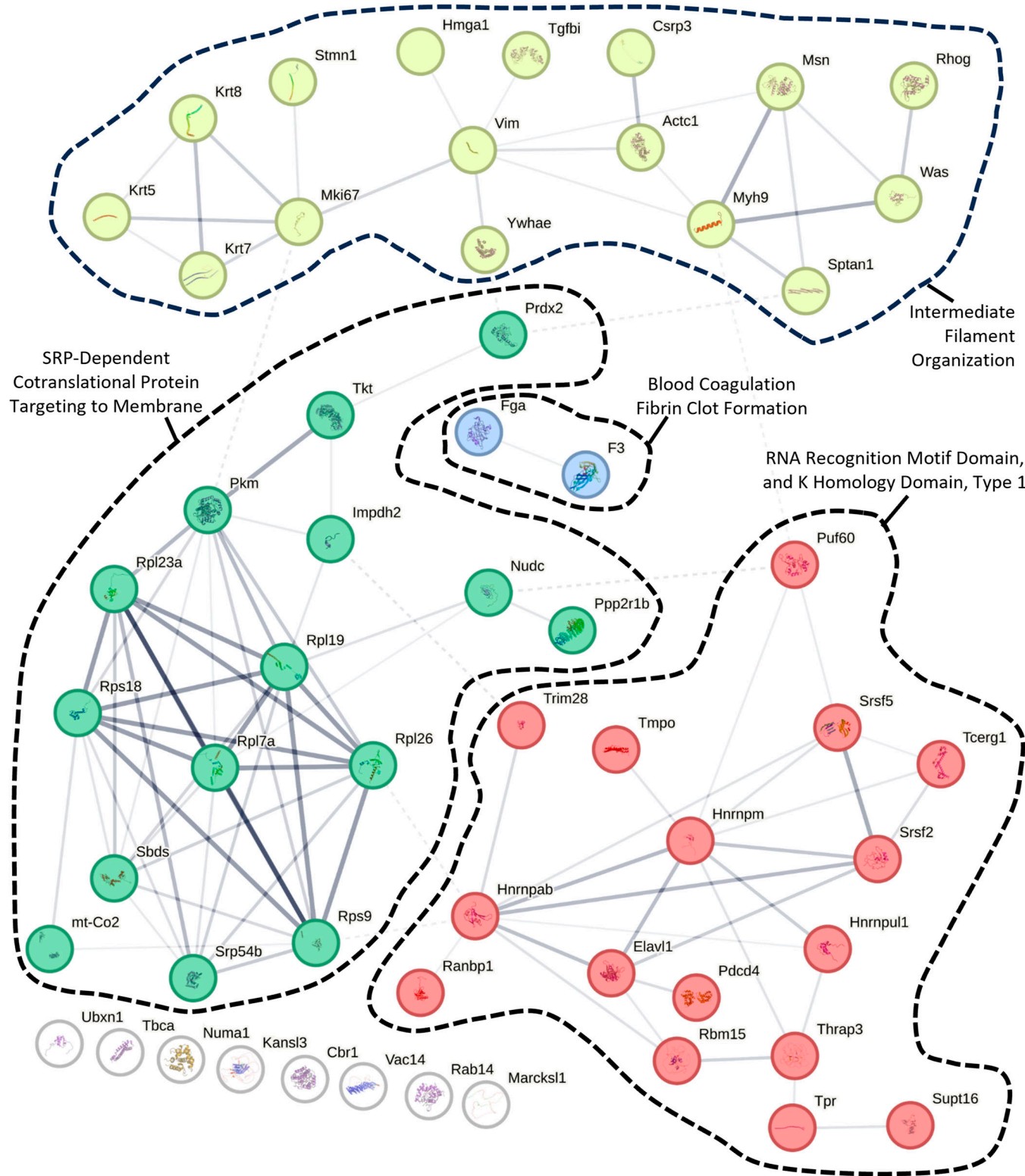

**Figure 5. Protein-protein interaction (PPI) analysis of 68 candidate proteins, identified through PepBridge in vehicle- and chemotherapy-treated murine thymi.**
The network was constructed using STRING v12.0 (Szklarczyk et al, 2023). Proteins were organized into distinct color-coded clusters using an in-built feature for k-means clustering. Associations between proteins are visualized using the medium confidence (0.4) modality.

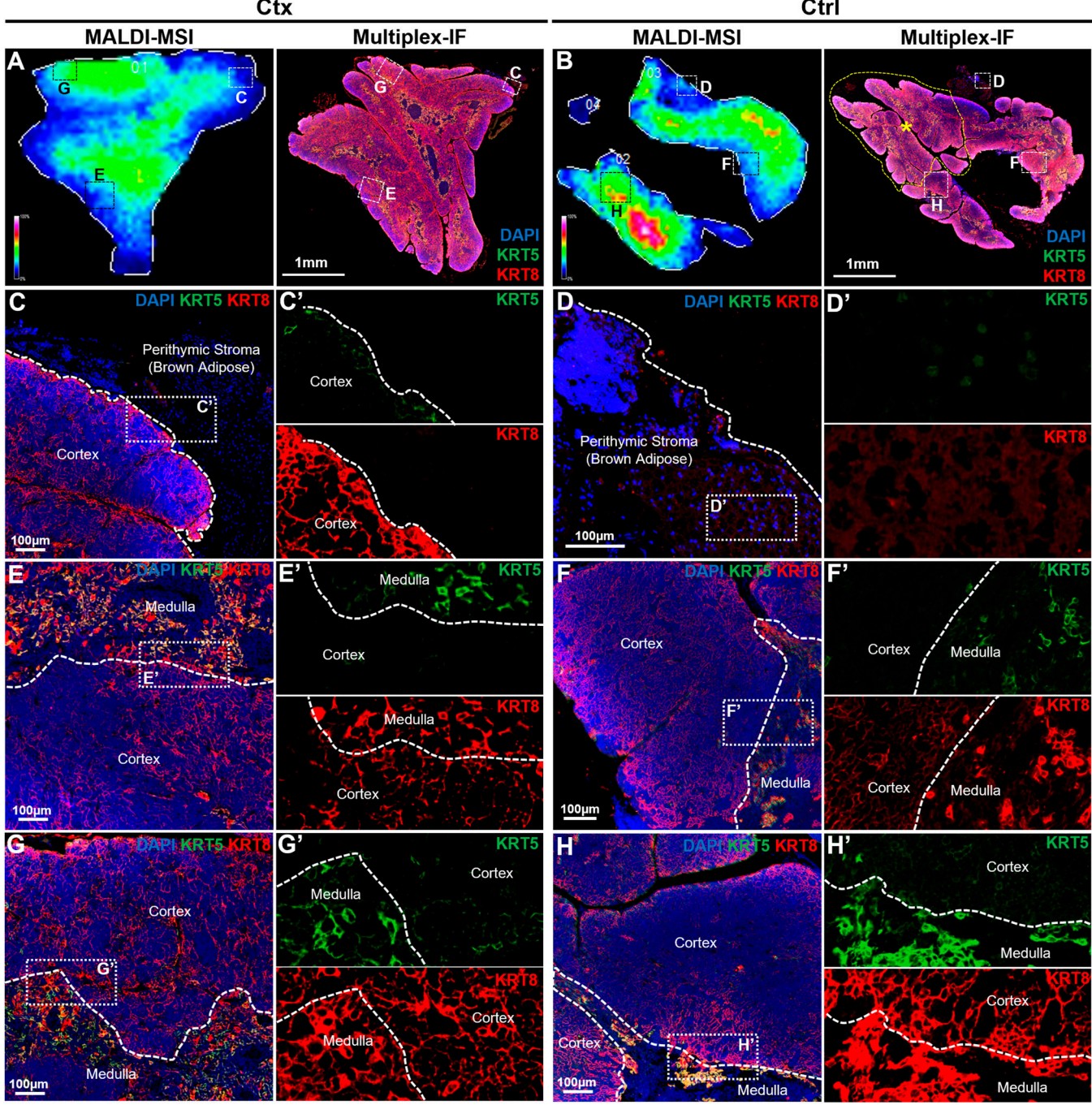

**Figure 6. Correlation between MALDI-MSI and KRT5/8 Immunofluorescence.**
**(A, B)** KRT8 (AEAETMoxYQIK, 1199.39 m/z) and KRT5 (FVSTTSSSR, 971.79 m/z) signal constructed from MALDI-MSI molecular images (left) along with their paired KRT5 and KRT8 immunofluorescence (right) on the sequential thymic sections of a cyclophosphamide-treated mouse (A) or a vehicle-treated mouse (B). Co-registration reveals mild shifts in the morphology and topography of the tissue, although most compartments can be captured. The yellow-dotted segments shown in the Ctrl animal on the right have been detached during the preparation and the processing of the MALDI-MSI and LC-MS/MS pipelines, and as such do not correspond to any segment on the MALDI-MSI acquisition. Impressively, the remaining component can be co-registered perfectly with the MALDI-MSI acquisition. **(C, D, E, F, G, H)** Small-dotted boxes corresponding to the letters (C, D, E, F, G, H), demonstrate the same fields-of-view in the MALDI-MSI acquisitions and the corresponding multiplex-IF sequential slides. **(A, B, C, D, E, F, G, H)** Magnified inserts of the six fields-of-view indicated in the dotted boxes in (A, B), assessed by multiplexed immunofluorescence, for DAPI (nuclear stain), keratin 5 (KRT5), and keratin 8 (KRT8). **(C, E, G)** correspond to the three fields-of-view in the cyclophosphamide (Ctx)-treated animal. **(D, F, H)** correspond to the three fields-of-view in the vehicle (Ctrl)-treated animal. **(C', D', E', F', G', H')** Hyphenated versions of these images, i.e., (C', D', E', F', G', H') represent magnified inserts of the white-dotted boxes in each corresponding figure, revealing in high-resolution and single-channels (top, KRT5; bottom, KRT8) the cellular details from their parental images. White-dotted lines correspond to the corticomedullary boundaries, as confirmed by KRT5 signal distribution and intensity, with the medulla staining as KRT5^high and other compartments as KRT5^− or KRT5^low.

the accuracy of PepBridge assignments, we validated additional proteins spanning distinct functional categories. These included nucleoprotein TPR (TPR) and tubulin-folding cofactor A (TBCA; identified in a subset of samples). For each, serial FFPE sections 5 μm apart from those used for MALDI-MSI were subjected to multiplex IF, enabling direct comparison of molecular images and histopathology (Fig 7). TPR, a nuclear basket protein of the nuclear pore complex, was consistently detected in high-resolution tim-sTOF flex datasets. MALDI-MSI mapped TPR predominantly to medullary regions with lower cortical abundance in both vehicle- and cyclophosphamide-treated thymi (Fig 7A–B'). This distribution was confirmed by immunofluorescence, with nuclear-localized signal enriched in the medulla (Fig 7C–D'). TBCA, a tubulin-folding chaperone important for microtubule assembly, was detected by MALDI-MSI in a subset of samples (Fig 7E and E'). When present, TBCA showed higher medullary abundance, confirmed by immunofluorescence (Fig 7F–H'). The variability of TBCA detection across animals may reflect differences in regeneration stage or protein abundance relative to MALDI-MSI sensitivity limits. Together, these results extend our validation beyond KRT5/8, to include nuclear pore components and cytoskeletal regulators, thus demonstrating the robustness of PepBridge assignments and the ability of MALDI-MSI to capture diverse subcellular localizations and expression patterns across thymic compartments.

## Discussion

This study introduces a pipeline for identifying spatially resolved protein biomarkers in the thymus by integrating MALDI-MSI and LC-MS/MS proteomics. By leveraging the complementary strengths of imaging and proteomics, we successfully mapped protein distribution across distinct thymic compartments and identified biomarkers associated with endogenous thymic regeneration after chemotherapy-induced involution. Our findings reveal significant changes in protein expression patterns within the thymic cortex and medulla during rebound hyperplasia. Key markers, including KRT8, TBCA, and TPR, exhibited distinct spatial distributions and dynamic expression profiles, reflecting critical processes involved in thymic remodeling after chemotherapy insult (Figs 6 and 7). Of note, KRT8, a prominent intermediate filament protein, which is well-established to be expressed in both cortical and medullary thymic epithelial cells (cTECs and mTECs) (Lee et al, 2011; Gupta et al, 2016), displayed expression in less defined corticomedullary boundaries in cyclophosphamide-treated mice (Ctx), potentially highlighting defective or ongoing remodeling.

Furthermore, our pipeline identified novel markers, such as TBCA, a tubulin-specific chaperone (Cowan & Lewis, 2001), which exhibited increased cortical expression during rebound hyperplasia, suggesting a pivotal role for cytoskeletal remodeling in thymocyte proliferation and cortical expansion. Similarly, TPR, a nuclear pore complex protein (Krull et al, 2010), demonstrated medullary-specific expression under control conditions but was significantly reduced after cyclophosphamide treatment, particularly in medullary thymocytes, emphasizing their vulnerability to chemotherapeutic insults. These findings highlight the potential of TBCA and TPR, as key markers for thymic compartmentalization

and regeneration. Whereas several top-scoring assignments were validated via immunofluorescence (e.g., KRT8, TPR, TBCA), this method may also probe false0positive candidates. Beyond identifying previously underrecognized factors of thymic remodeling, this study is the first to present a spatiotemporally characterized proteome of thymic compartmentalization and recovery. This innovative combination enabled high-confidence protein mapping across thymic compartments, yielding unprecedented spatial resolution of proteomic changes. By leveraging the strengths of both technologies, this approach provides not only a detailed atlas of thymic proteins but also a framework for future studies to explore complex tissue dynamics with enhanced precision and reliability. These advancements position this study as a significant methodological leap forward in understanding thymic regeneration.

Immune and thymic recovery post-chemotherapy is a critical clinical challenge, particularly in pediatric cancer patients, who often experience long-term immune dysfunction because of chemotherapy-induced thymic involution. The thymus plays a central role in T-cell development, and its damage can lead to impaired thymopoiesis, diminished T-cell output, and compromised immune reconstitution. This creates a vulnerability to infections, secondary malignancies, and inadequate immune surveillance, which are significant contributors to morbidity and mortality in pediatric cancer survivors (Lagou et al, 2022). Studies such as this one address this critical need by providing foundational insights into molecular mechanisms of thymic regeneration, offering potential therapeutic targets and/or functional biomarkers of thymopoiesis. Whereas it is unlikely that MALDI-MSI coupled with LC-MS/MS will be directly implemented in clinical care for pediatric cancer patients (given that thymic biopsy is highly invasive and not part of standard management), this technology holds great promise in preclinical settings. It could be used to establish pharmacodynamic biomarkers to monitor immune recovery during cytotoxic treatments in preclinical mouse models (Califf, 2018), thereby enhancing the discovery of novel drugs or drug combinations aimed at achieving immunological/thymic recovery post-chemotherapy. For example, proteins such as TBCA and TPR identified in this study and implicated in cytoskeletal remodeling, compartmental stability, and thymic regeneration, could serve as biomarkers for monitoring thymic recovery after cyclophosphamide treatment in animal research. Cyclophosphamide remains a cornerstone chemotherapeutic agent in the management of pediatric malignancies (Gilheeney et al, 2007; Yanagisawa et al, 2009; El Kababri et al, 2020), and leveraging such biomarkers in preclinical models could provide critical insights to optimize therapeutic strategies and improve long-term immune outcomes for pediatric patients.

Our spatial proteomics platform is broadly applicable beyond thymic mapping. By integrating MALDI-MSI with LC-MS/MS, it enables spatially resolved protein identification in lymphoid organs such as lymph nodes, spleen, thymus, and bone marrow, key sites for immune surveillance, pathogen defense, and hematopoiesis. Mapping proteomes in these tissues can reveal biomarkers and pathways that mediate immune interactions in health and disease. Prior MALDI-MSI studies have indeed mapped immune mediators

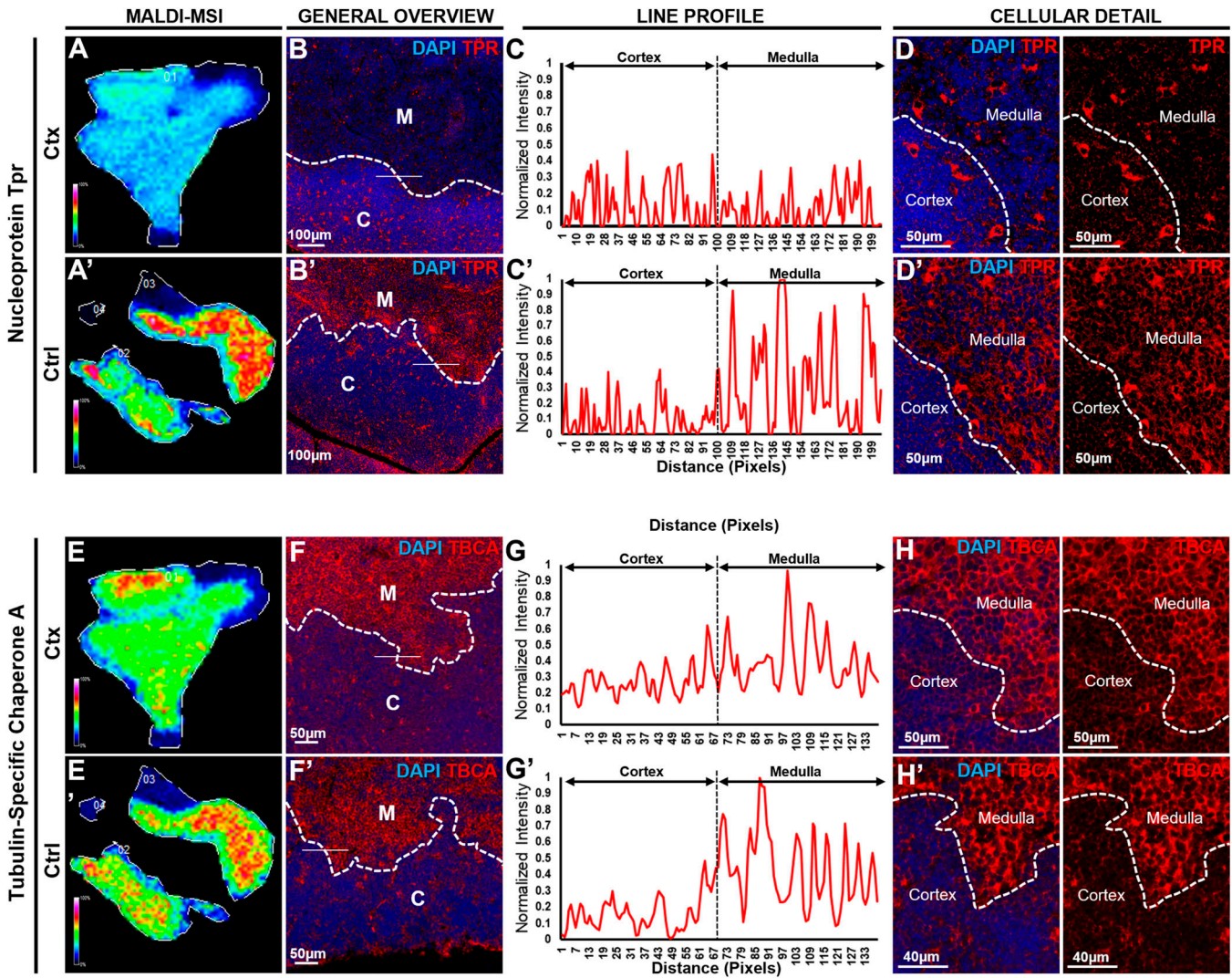

**Figure 7. Immunolocalization of novel biomarkers of thymic compartmentalization in murine thymi at cyclophosphamide-induced rebound hyperplasia.**
**(A, B, C, D, E, F, G, H)** Validation of two novel markers identified through our new MALDI-MSI pipeline, nucleoprotein Tpr (A, B, C, D), tubulin-specific chaperone A (TBCA) (E, F, G, H). For each immunolabelled protein, a series of sequential representations is reported in a similar manner. The top rows represent cyclophosphamide (Ctx)-treated mice, whereas the bottom rows (hyphenated versions) represent vehicle (Ctrl)-treated mice. **(A, A', E, E')** The first column reveals the molecular image of the respective protein, as obtained by MALDI-MSI acquisition, in this case, TPR (ESLLAEQR, 946.03 m/z) (A, A') and TBCA (LEAAYTDLQQILESEK, 1850.88 m/z) (E, E'). **(B, B', F, F')** The second column reveals the immunolocalization of each respective protein and condition, using multiplex-immunofluorescence in paraffin-embedded tissues, and fields-of-view are representative of at least N = 3, in this case, TPR (B, B') and TBCA (F, F'). In all cases, DAPI was used as nuclear stain, to document the presence of tissue, and demarcate the corticomedullary junction (white-dotted lines). C, cortex; M, medulla; white-dotted lines indicate corticomedullary junctions. **(B, B', F, F')** In each image a white line (206 pixels for (B, B'), and 137 pixels for (F, F')) crosses through the corticomedullary junctions and is used for line profiling in the next column. **(C, C', G, G')** In the third column, the normalized signal intensity of the corresponding protein is plotted along the line profile, in this case, TPR (C, C') and TBCA (G, G'). The black dotted lines indicate the pixel which segments the line into its cortical (left) and medullary (right) side. Intensity values are normalized to the maximum intensity observed from the two line profiles of the same immunolabelled protein. The line profiles are representations of at least N = 20 lines measured in each condition. The fourth and fifth columns represent higher magnifications of the region around the line profiles of the second column. **(D, D', H, H')** They reveal at cellular detail the precise immunolocalization of each respective protein and condition, using multiplex-immunofluorescence in paraffin-embedded tissues, and fields-of-view are representative of at least N = 10, in this case, TPR (D, D'), and TBCA (H, H'). For each example, the fourth column (left) is a merged image including DAPI as a nuclear stain, whereas the fifth column (right) is a single channel of the corresponding immunolabelled protein. White-dotted lines indicate the corticomedullary junction.

in infected lymph nodes (Do et al, 2020), identified metastasis markers in cancer-draining nodes (Mittal et al, 2016), and profiled lipid changes in injured spleens (Xie et al, 2024). However, high-resolution protein mapping in complex architectures remains challenging. Our combined approach addresses this by providing confident protein IDs with spatial context, enabling detailed

proteomic atlases under conditions such as infection, autoimmunity, and cancer.

PepBridge is designed for widely available MALDI-TOF/TOF instruments, offering high throughput for labs without ultra-high–resolution MSI, when also working with high-resolution data from platforms such as the timsTOF flex. It uses a scoring-

based strategy that integrates mass accuracy, abundance changes, and intensity ranking to rapidly match MALDI peaks with LC-MS/MS peptides, narrowing candidates for further validation. PepBridge is not a replacement for high-resolution workflows such as HIT-MAP but a bridge to link MALDI-MSI and LC-MS/MS datasets. Protein IDs in our pipeline require at least one confidently matched tryptic peptide, less stringent than conventional proteomics but necessary given the limited peptide coverage of on-tissue digestion. Multiple matched peptides, when present, increase confidence. Current limitations include the lack of formal FDR estimation, as the algorithm always returns a top match. Future versions will integrate decoy-based confidence scoring. Until then, reliability depends on meeting all scoring criteria, which in our tests have been evaluated on a limited but representative dataset.

In conclusion, this study establishes a spatial proteomics pipeline that not only advances our understanding of thymic regeneration but also lays the foundation for broader applications across lymphoid organs. By combining MALDI-MSI and LC-MS/MS with high-confidence protein mapping, this platform offers unparalleled insights into tissue compartmentalization and immune interactions, with the potential to inform therapeutic strategies and improve immune recovery in diverse pathological contexts.

## Materials and Methods

### Mouse model of cyclophosphamide-induced thymic involution

FVB/NCrl female mice (n = 5), 7–8 wk old were used for this study. Cyclophosphamide monohydrate (CTX, Thermo Fisher Scientific) was reconstituted at a concentration of 25 mg/ml in sterile PBS. Each mouse in the experimental group received an *i.p.* dose of 200 mg/Kg of cyclophosphamide monohydrate in sterile PBS (200 $\mu$l total volume), every 3 d, for a total of three doses. The control (vehicle-treated) mouse group received an *i.p.* injection of 200 $\mu$l sterile PBS. The last day of chemotherapy treatment was designated as "day 0." The mice were then blindly categorized into five subgroups. Mice from either control or experimental subgroups were euthanized on a weekly basis for up to 5 wk post-chemotherapy, i.e., day 35. Mouse thymi were dissected using standardized autopsy procedures, and were cleaned from perithymic adipose tissue, carefully without damaging the thymus. Experiments were replicated in at least three independent mouse cohorts (N = 5 mice per group).

### MALDI-MSI analysis of mouse thymus tissue sections

Thymi from vehicle- and cyclophosphamide (Ctx)-treated mice were fixed in 10% neutral buffered formalin for 48 h, processed, and paraffin embedded (Fig 1A). FFPE thymic tissue sections, cut at 5-$\mu$m thickness, were mounted on indium-tin oxide (ITO) conductive glass slides. The FFPE tissue sections were deparaffinized for 1.5 h at 60–65°C and washed through a series of xylene, 100%, 95%, 70% ethanol and water. Tissue slides in the 10 mM Tris buffer were heated to 95–98°C using a steamer for 30 min. Slides were cooled down, buffer exchanged with water, and dried in a

desiccator overnight. The tissue slides were then spray coated with trypsin solution using the TM Sprayer (HTX Technologies). Trypsin (0.05 $\mu$g/$\mu$l; in 50 mM ammonium bicarbonate/10% acetonitrile) was sprayed on the tissue slide using the following parameters: nozzle temperature (30°C), 10 passes, 0.025 mg/ml flow rate, 750 mm/min nozzle velocity, 2 mm track spacing, HH spray pattern, nitrogen gas pressure set at 10, nitrogen gas flow rate (3 liters/min), 0 s drying time, 40 mm nozzle height. The total amount of trypsin was 20 $\mu$g per slide. The slides were air dried for 5 min and then incubated in a hydrated chamber for overnight trypsin digestion at 37°C. After digestion, slides were dried for 15 min in the vacuum desiccator and then spray coated with CHCA matrix ($\alpha$-cyano-hydroxycinnamic acid; 5 mg/ml in 50% AcN/0.2% TFA) using the TM Sprayer (HTX technologies) with the following settings: nozzle temperature (75°C), 4 passes, 0.1 ml/min flow rate, 1,300 mm/min nozzle velocity, 2 mm track spacing, CC spray pattern, nitrogen gas pressure set at 10, nitrogen gas flow rate (3 liters/min), 10 s drying time, 40 mm nozzle height. The final matrix density was 0.00077 mg/mm$^2$ or 0.8 $\mu$g/mm$^2$.

Peptide spatial mapping was performed by scanning and acquiring spectra m/z range (700–3,500) at 100 $\mu$m raster width and 800 laser shots/pixel using the UltrafleXtreme MALDI-TOF/TOF mass spectrometer (Bruker Daltonics). Instrument parameters were set as follows: partial raandom walk, shots at raster spot: 49, diameter limit: 2,000 $\mu$m, feflector gain: 2,414 V, smart beam parameter: small, frequency: 1,000 Hz, sample rate and digitizer: 1.00 Gs/s, ion source 1: 24.97 kV, ion source 2: 22.32 kV, lens: 7.41 kV, reflector: 26.54; reflector 2: 13.41 kV, matrix suppression: deflection, processing: SC peptide cent. After the MALDI imaging analysis, the matrix was washed off from the tissue and followed by staining with (H&E). Spectra were processed by pre-defined smoothing and baseline subtraction. To remove variations in pixel-to-pixel intensities, TIC (mean) normalization was performed by dividing all spectra by the means of all data points. Using Fleximaging v 4.0, unsupervised hierarchical clustering based on the Euclidean distance metric and Ward linkage was performed to generate segmentation maps from the images acquired from the tissue samples. The dendrogram was generated with only the top nodes displayed. To delve into additional layers into the dendrogram, the "+" sign was selected to expand the tree. The numbers in brackets corresponded to the number of mass spectra belonging to the same node, whereas the number before it corresponded to the node ID or the cluster distance. The selected branch generated a display filter assigned to a specific color for clustering pixels with similar spectra. This generated a spatial heatmap that uses a color intensity scale to show the varying intensity ratios of mass signals observed from different areas on the tissue. The initial results obtained from the UltrafleXtreme were further assessed by analyzing additional tissue sections from different time points from post-cyclophosphamide treatment using the timsTOF flex mass spectrometer.

### Spatial proteomics

Peptide spatial mapping was performed by scanning and acquiring spectra m/z range of 700–3,500 at 50 $\mu$m raster width timsTOF flex mass spectrometer (Bruker Daltonics) equipped with a MALDI-2

laser and trapped ion mobility (TIM). Instrument parameters were set as follows (For these experiments, the TIMS mode was active, whereas the MALDI-2 post-ionization source was not engaged): General: MALDI Plate Offset: 50 V, Deflection 1 Delta: 70 V, Funnel 1 RF: 500 Vpp, isCID Energy: 0 V, Funnel 2 RF: 350 V, Multipole RF: 400 V, Collision Energy: 20 eV, Collision RF: 2,500 Vpp, Quadrupole Ion Energy: 10 eV, Low Mass: 300 m/z, Transfer Time: 110 $\mu$s, Pre Pulse Storage: 10 $\mu$s. TIMS: deltaT1: −20 V, deltaT2: −120 V, deltaT3: 70 V, deltaT4: 100 V, deltaT5: 0 V, deltaT6: 200 V, Collision Cell In 220 V. The timsTOF flex use the SciLS Lab software for data processing to generate peptide ion maps. For timsTOF fleX MALDI-2 data, several built-in functions were used to visualize data, including segmentation. To perform segmentation analysis, a feature list was generated by taking the 10,000 most frequent features across all pixels in the entire m/z range, with the most frequent features being present in most pixels, regardless of relative intensity. Features were determined upon importation of the data into SCiLS Lab, where the raw data are binned into features based on the default SCiLS binning parameters, and can be thought of as m/z picked peaks. Segmentation was performed by clustering pixels, using the mass spectra of each pixel across the entire m/z range, using only the features found in the feature list mentioned. The clustering was performed using a bisecting k-means algorithm that uses correlation distance as a metric, with each pixel spectrum, both denoised using the weak denoising setting and normalized against the total ion count of the respective pixel. Further statistical analysis was performed using SCiLS Lab's built-in 10 principal component analysis using the same 10,000 feature list, normalizing each pixel to the total ion count and scaling the relative intensities of the features to unit variance.

### LC-MS/MS and protein identification

Serial or adjacent FFPE tissue sections at 5 $\mu$m thickness were mounted on positively charged regular microscope glass slides (Fig 1B). The FFPE tissue sections were deparaffinized for 1.5 h at 60–65°C and washed through a series of xylene, 100%, 95%, 70% ethanol, and water. Tissue slides in the 10 mM Tris buffer were heated to 95–98°C using a steamer for 30 min. Slides were cooled down, buffer exchanged with water and dried in a desiccator overnight. The tissue slides were then spray coated with trypsin solution using the TM Sprayer (HTX Technologies). The slides were air dried for 5 min and then incubated in a hydrated chamber for overnight trypsin digestion at 37°C. After digestion, the slides were dried for 15 min in the vacuum desiccator. The peptides were sequentially extracted twice each by pipetting 30 $\mu$l of 0.1% triflouroacetic acid and 60% acetonitrile/0.1 triflouroacetic acid, collected and pooled into Eppendorf microcentrifuge tube.

The peptide digests were desalted using C-18 ziptip or Oasis HLB resin by vacuum filtration and eluted with 60% acetonitrile/0.1% TFA, concentrated in a speed vacuum and re-suspended in 0.1% acetonitrile and analyzed by LC-MS/MS using a nanoRSLC U3000 (Dionex, Thermo Fisher Scientific) or a nanoAcquity UPLC (Waters) connected to Orbitrap Exploris 480 (Thermo Fisher Scientific). The LC gradient is composed of 120 min for 4–34% solvent B (solvent A: 0.1% formic acid and solvent B: 80% acetonitrile/0.1% formic acid).

For peptide identification and label-free quantification, the MS/MS data obtained from DDA were searched by the SEQUEST search engine using Proteome Discoverer v2.5 (Thermo Fisher Scientific). Protein abundances were transformed into $log_2$ and normalized, and statistical differences between relative protein abundances were assessed using a $t$ test as previously discussed (Aguilan et al, 2020). Automated database search was performed by Proteome Discoverer v2.5 using Sequest search engine. MS/MS data obtained from DDA searched against the SwissProt mouse database with the following parameters: missed cleavage = 2; oxidation on M; deamidation on N and Q; peptide tolerance = 10 ppm; MS/MS tolerance = 0.02 Da for MS/MS spectra acquired in the orbitrap analyzer.

### PepBridge: the MALDI-MSI and LC-MS/MS peak-matching pipeline

For the data obtained from the UltrafleXtreme, data obtained from the average spectrum was first processed by reducing the bin size to 4,000 and then converted to a .csv file containing m/z versus MALDI average intensities. On the other hand, the .csv file from the timsTOF flex was converted to an Excel file and the exported peptide table obtained from the LC-MS/MS analysis was pasted into the same file. The peptide table was sorted by the highest intensity of MALDI peaks and further processed by an in-house developed program (pepBridge.py) described in detail in Supplementary Materials and Methods section and Fig S3, which can be downloaded from this link: https://github.com/carlos-madrid-aliste/pepBridge. An m/z with a filter of ±1.0 Da was used for both the UltrafleXtreme and timsTOF flex, to match the MALDI m/z with the corresponding theoretical $[M+H]^+$ for each peptide identified from the LC-MS/MS analysis. The MALDI average intensities were then sorted from highest to lowest and plotted. MALDI-LC/MS/MS matches with MALDI average intensities falling below the inflection point of the curve (an arbitrarily assigned noise cut-off filter), were filtered out. Both MALDI average intensities and LC-MS/MS intensities were $log_2$ transformed. Each $log_2$ transformed peptide was normalized by the average intensities of all the peptides identified per sample. MALDI-TOF suffers from low resolution and peak widths are broader, when compared with the narrower peaks obtained from high resolution mass spectrometers used for LC-MS/MS, such as the orbitrap. A single peptide peak observed from MALDI-TOF with corresponding peptide ion heat map can potentially match to ≥1 peptide candidate observed from LC-MS/MS. Hence, we validated the results we obtained from the UltrafleXtreme using the timsTOF flex mass spectrometer. To provide a basis for ranking of top peptide candidates based on probabilities, we developed a MALDI-LC-MS/MS peak-matching pipeline, which served as a (prioritization) scoring system, based on four (4) metrics.

### Metric 1: delta of $log_2$ ratios or $log_2$ fold changes between MALDI and LC-MS/MS

The differences between the $log_2$ fold changes of Ctx versus Ctrl between MALDI and LC-MS/MS were calculated and sorted from lowest to highest and plotted against the scores. The highest score, y = 10 corresponded to the smallest delta or difference between the $log_2$ fold changes of "Ctx" versus "Ctrl" obtained from MALDI and LC-MS/MS. Hence, a Gaussian curve was generated with the x

centered at 0 $log_2$ fold change versus the y maximum of 10. The formula is $f(x) = A \cdot e^{-[(x-\mu)^2/2\sigma^2]}$ where x is the input variable, A is the amplitude, which is the maximum value of the distribution, $\mu$ is the mean (or the center) of the distribution, and $\sigma$ is the SD that controls the spread of distribution. Because A was assigned to be the highest score of 10 and x = 0 to be at the center, the formula becomes $f(x) = 10 \cdot e^{-[(x)^2/2\sigma^2]}$. This was converted to an Excel formula, y = 10*EXP(−A1^2/(2*(B1^2))), where A1 represents the cell containing the input value of x (delta $log_2$ fold change), and B1 represents the SD derived from all the delta $log_2$ fold changes. This formula calculates the y values based on a Gaussian distribution using the specified parameters.

### Metric 2: delta mass between MALDI and LC-MS/MS

The differences between the peptide masses observed from MALDI and LC-MS/MS were calculated. The highest score of y = 100 corresponds to the x with the smallest difference as x approaches 0, whereas the minimum score of y = 0 corresponds to x with the biggest peptide mass difference observed from MALDI and LC-MS/MS as x approaches 1. The function would be $f(x) = 1/1 + e^{k(x-x0)}$ where x is the computed delta mass, x0 is the central value where x = 0.5 and k is the steepness parameter. In Excel, the formula is y = 100/(1 + EXP(10*(A1 − 0.5))), where A1 is the computed delta mass, the steepness parameter, k = 10 and central value x = 0.5. This formula calculates the y values based on the negative sigmoid function with the central value x = 0.5.

### Metric 3: MALDI intensity

The third metric is the MALDI intensities. The higher intensity means the higher score where y = 5 as the maximum score and y = 0 is the minimum score versus x which are the MALDI intensities with a central value of x = 2. So, the function is, $f(x) = 5/1 + e^{-k(x-2)}$, where y = 5 is the maximum score, k is the steepness parameter, and x is the MALDI intensity. In Excel, the formula is, y = 5/(1 + EXP(−A1 + 2)), where A1 is the input value, which is the MALDI intensity and the steepness parameter, k = 1. This formula calculates the y values based on the sigmoid function using the specified parameters.

### Metric 4: matching with the image or the peptide ion heat map

The scores from Metrics 1–3 are summed, with the highest total representing the top candidate peptide, followed by the other candidate peptides corresponding to a specific MALDI peak (m/z). Metric 4 serves as the final criterion, aligning the peptide's spatial mapping or MALDI molecular image with both the log-fold changes and peptide scores.

## Bioinformatic characterization of prioritized candidates

The protein accession numbers of the prioritized candidate proteins were submitted to STRING v12.0 (Szklarczyk et al, 2023), for functional network association analysis. The network was visualized by confidence meaning of network at medium confidence (0.4), and all active interaction resources were included in the analysis. The k-means clustering approach was selected for classifying the protein subnetworks into thematic clusters involving distinct biological/molecular concepts.

## PepBridge.py program description

PepBridge.py is a Python program designed to map MALDI-identified m/z values with LCMS m/z values. The inspiration for this program came from our efforts to match MALDI with LCMS m/z values using Excel's MATCH function. This function is typically used to determine the position of a lookup value within a column table, returning its relative position. However, the MATCH function in Excel has a limitation—it only returns the position of the first exact match encountered.

PepBridge.py overcomes this limitation by seamlessly associating MALDI-identified m/z values with their corresponding LCMS values within a user-defined threshold. At its core, PepBridge.py reads two CSV files containing MALDI and LCMS m/z values and generates a consolidated file containing all LCMS rows matching the MALDI m/z value.

The program logic is straightforward: it begins by loading MALDI and LCMS data files in CSV format, extracting the respective masses. Initially stored as Python lists, these masses are later converted to NumPy arrays for enhanced search efficiency. After extracting the masses from both LCMS and MALDI data files, the program iterates over the MALDI masses, identifying matches in the LCMS data using the get_hits method. This method effectively retrieves all LCMS matching masses within a specified threshold. To ensure a well-organized structure, the program adopts an object-oriented approach. PepBridge.py comprises three main classes: MatchingMZ.py, MALDI_LCMS.py, and Config_file.py.

Config_file.py primarily serves as a class for reading and manipulating configuration files in INI format. It offers functionality and methods for reading INI files, allowing users to specify various parameters such as the location of LCMS or MALDI output files, whether m/z values should be rounded to the nearest integer before matching, and which columns should be read from the files (such as m/z, average, control, and experimental values). This flexibility is invaluable, particularly when dealing with CSV containing numerous columns. Furthermore, the [ordered_columns] field within the INI file enables users to specify the desired order in which columns should be printed, adding an extra layer of customization to the output.

MALDI_LCMS is the class that triggers the reading of INI configuration files and initializing the corresponding LCMS and MALDI objects. It loads the MALDI/LCMS CSV data into memory and provides functionality to order the file columns according to the user's specified order and to extract the columns holding the MALDI m/z columns (rounded or not) used for a precise or approximate matching among the LCMS masses (m/z).

The MatchingMZ class serves as the workhorse, performing the LC-MS and MALDI class initialization and providing functionality to perform the m/z mapping between MALDI to LC-MS data and store the matches into a file.

## Immunofluorescence (IF)

5 $\mu$m sections, serial to the ones used for MALDI-MSI analysis, were deparaffinized and dehydrated in double changes of xylene and 100°, 95°, 70° alcohol, respectively, followed by rehydration. Citrate (Novus Biologicals) and EDTA (Epredia) antigen retrieval was performed with a steamer for 21 min. The blocking step was performed

with 5% goat serum (R&D systems) in 0.05% PBS-Tween20 for 1 h, after which slides were incubated with the following primary antibodies overnight at 4°C: TBCA (Rabbit, polyclonal, Invitrogen), TPR (Rabbit, polyclonal, Invitrogen), KRT5 (Chicken IgY, purified polyclonal, Biologend), KRT8 (Guinea Pig, polyclonal, Origene). All antibodies were used at 1:100, except KRT5, which was used at 1:300. The secondary antibody incubation was performed with the following antibodies at 1:200 dilution for 50 min: goat anti-Rabbit Alexa Fluor488 (Invitrogen), goat anti-Chicken IgY Alexa Fluor488 (Invitrogen), goat anti-Guinea Pig Alexa Fluor555 (Invitrogen), followed by nuclear staining for 5 min with 4,6-Diamidino-2-phenylindole dihydrochloride (DAPI, Novus Biologicals) and mounting with ProLong Gold anti-fade mounting reagent (Invitrogen).

Slides were left to dry overnight at dark, and subjected to digital scanning via a 3D HISTECH P250 Flash III automated slide scanner, using a 20x 0.75NA objective lens into a multichannel overlay image, as described in prior work from our group (Karagiannis et al, 2017; Borriello et al, 2020; Coste et al, 2020; Sharma et al, 2021). All used slides were subjected to the same color adjustment for each marker in CaseViewer version2.4 software.

Sequential tissue sections analyzed by MALDI-MSI and IF were digitally co-registered using established whole slide imaging (WSI) methodologies, including tissue-based landmarks and fiducial markings, to ensure accurate spatial alignment (Shafique et al, 2021; Doyle et al, 2023), adapted to our setting. Registration was performed by matching distinct morphological and anatomical features (e.g., vasculature, epithelial boundaries, and stromal contours) across the two different imaging modalities.

### Histopathology

H&E staining was conducted with the same deparaffinization procedure as described above, which was then followed by Hematoxylin incubation for 5 min, differentiation for 5 s, Eosin Y stain for 1 min 45 s, clearing in 95, then 100° alcohol, and then xylene, and finally mounting with Permount (Thermo Fisher Scientific).

## Data Availability

The mass spectrometry data have been deposited to the ProteomeXchange Consortium via the PRIDE partner repository with the dataset identifier PXD058296.

## Supplementary Information

## Acknowledgements

The Karagiannis Lab gratefully acknowledges the Montefiore-Einstein Comprehensive Cancer Center for funding through its support grant (P30 CA013330), the Analytical Imaging Facility and its dedicated personnel for the use of the Fast Scanner, acquired via shared instrumentation grants (1S10 OD026852-01A1), and the Rally Foundation for the Young Investigator Award to GS Karagiannis. The Sidoli Lab gratefully acknowledges funding from the Hevolution Foundation (AFAR), the Einstein-Mount Sinai Diabetes Center, and the NIH Office of the Director (S10 OD030286).

### Author Contributions

JT Aguilan: data curation, software, formal analysis, visualization, methodology, and writing—original draft.
C Madrid-Aliste: data curation, software, formal analysis, and methodology.
J Fischer: resources, formal analysis, visualization, and methodology.
MK Lagou: resources.
S Sidoli: conceptualization, resources, formal analysis, supervision, investigation, methodology, and writing—review and editing.
GS Karagiannis: conceptualization, formal analysis, supervision, investigation, visualization, methodology, and writing—original draft, review, and editing.

### Conflict of Interest Statement

The authors declare that they have no conflict of interest.

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
