## [Reviewer comments · Life Science Alliance]

Identifying Space-Resolved Proteins of the Murine Thymus, by Combining MALDI-MSI and Proteomics

Jennifer Aguilan, Carlos Madrid-Aliste, Joshua Fischer, Simone Sidoli, and George Karagiannis

DOI: <https://doi.org/10.26508/lsa.202503205>

Corresponding author(s): George Karagiannis, Albert Einstein College of Medicine and Simone Sidoli, Albert Einstein College of Medicine

Review Timeline:

Submission Date:	2025-01-06
Editorial Decision:	2025-02-13
Revision Received:	2025-09-10
Editorial Decision:	2025-10-08
Revision Received:	2025-10-17
Accepted:	2025-10-21

Scientific Editor: Tim Fessenden

Transaction Report:

February 13, 2025

Re: Life Science Alliance manuscript #LSA-2025-03205-T

Dr. George Karagiannis
Albert Einstein College of Medicine
Microbiology and Immunology
1300 Morris Park Avenue
Forchheimer Building, Room 640
Bronx, NY 10461

Dear Dr. Karagiannis,

Thank you for submitting your manuscript entitled "Identifying Space-Resolved Proteins of the Murine Thymus, by Combining MALDI Mass Spectrometry Imaging and Proteomics" to Life Science Alliance. The manuscript was assessed by expert reviewers, whose comments are appended to this letter. We invite you to submit a revised manuscript addressing the Reviewer comments.

Thank you for this interesting contribution to Life Science Alliance. We are looking forward to receiving your revised manuscript.

Sincerely,

B. MANUSCRIPT ORGANIZATION AND FORMATTING:

Reviewer #1 (Comments to the Authors (Required)):

In the presented study the authors propose an approach to identify proteins from MALDI imaging measurements of peptides. This is achieved by performing additional LC-MS/MS experiments on a separate tissue sections and combining the results via a new software method, termed "pepBridge". The applicability of the method is demonstrated on mouse model of chemotherapy-induced thymic involution.

I would like to begin my review with one of the sentences of the article at the end of the discussion:

"Although several on-tissue bottom-up proteomics studies have combined peptide spatial mapping by MALDI-MSI with LC-MS/MS to identify protein markers from tissue sections [120-126], there is still no streamlined method for directly correlating these datasets".

1) While I agree with the latter, with respect to the first part of the sentence, I would like to point out that 6 out of the 7 references #120-#126 refer to papers from the years 2011 or earlier. In the meantime, several approaches have been proposed based on instrumentation with higher mass resolution such as:

a) Schober et al. (DOI: 10.1002/rcm.5135) have already performed mass matching experiment from MALDI imaging to LC/MS experiments in 2011 and came to the conclusion that "All results (including MALDI imaging data) were based on accurate mass measurements (RMS <2 ppm) and allow a confident identification of tryptic peptides. Measurements based on lower accuracy would have led to ambiguous or misleading results". Aguilan et al. are using matching tolerances of up to 1 Dalton which corresponds to 500 ppm tolerance @mz 2000, hence 250 times higher tolerance (and hence uncertainty) than proposed by Schober 15 years ago. Hence recent papers use high resolution mass spectrometry, at least as intermediate between MALDI and ESI, to perform mass matching, e.g. Kip et al. (DOI: 10.1021/acs.jproteome.1c00447).

b) Guo et al. (DOI: 10.1038/s41467-021-23461-w) have proposed in 2021 a peptide mass fingerprint based approach (HIT-MAP) using also spatial segmentation of the data and a decoy databased to estimate false discoveries.

Hence it is not clear, where the approach presented in this paper is standing as compared to the developments in the field from 2011 onwards. The authors should hence compare their approach at least with the two above mentioned approaches and work out why their approach is better or complementary to the existing approaches!

2) As consequence my following concerns go to the pepBridge scoring system itself:

a) it is not clear why the distances/intensities or whatever in metrics 1, 2, and 3 need to be mapped to the exponential models as shown in in Figures 5CDE?

b) it is not convincing why the highest intensity in MALDI should obtain the highest score? What is the rational behind that?

c) for metric 1: what happens if there is no control group or if there is no control group in LC-MS/MS (e.g. when comparing ad hoc two regions in the MSI dataset)?

d) for metric 4: since there is no image for the LC-MS/MS data, how can this help in matching? The rational in the metric description is very vague and does not give answer to the question stated: "Metric 4 serves as the final criterion, aligning the peptide's spatial mapping or MALDI molecular image with both the log-fold changes and peptide scores"

e) most importantly, what is your probability of having a false alignment? Since there will be always a best hit (and there is no threshold), what is the chance that this has been an erroneous assignment? The authors should provide a strategy to evaluate before performing IHC/IF what the false discovery rate is.

f) in that light, has there been any protein assignment which could not be successfully validated with IF?

g) it is not clear, how protein identification is performed when actually peptide masses are matched. Is there a threshold, e.g. for a minimum number of peptides that need to be detected in the MSI dataset of the same protein, so that the protein can be considered detected? Why is the number of peptide matches not incorporated in the scoring system?

h) according to Table S2 the tolerance of 1 Da is also creating duplicate assignments to the isotope signals in the LC-MS/MS

data (e.g. histone H4: MALDI m/z 1326.06 is assigned to 1325.75 and 1326.74 of the same peptide sequence). Likewise, isotope signals in the MALDI MSI data (e.g. Histone H2 at 944 with isotope signals at 945 and 946) could lead to wrong matches in the LC-MS/MS data. Did the authors perform isotope removal in both datasets? If not, how do the authors see any way out of this? i) ultimately, the scoring system has not been validated extensively, but only on 1-2 proteins and their corresponding IF images. The authors should how their approach should be validated more comprehensively!

3) Related to the pepBridge approach, I have some questions on the experiments:

a) has the instrument's mass axis been calibrated before every experiment? If yes, how? What was the average mass accuracy error in ppm and standard deviation? This is an important parameter for a later mass matching.

b) the binning to 4000 data points (most likely the reduction factor is one order of magnitude) will again lead to a reduction in mass accuracy. Why did the authors do that? Also, can the authors calculate the loss in mass accuracy due to the loss of mass resolution?

4) Minor comments:

a) Introduction: "However, MALDI-MSI is generally limited due to the low signal-to-noise ratio² and the limited capability of performing MS/MS fragmentation of candidates, which reduces the specificity of detection³." This is an oversimplified description of the real limitations which are 1) the lack of chromatographic separation, leading to dominance of most abundant proteins in the measurements; 2) the use of a linear TOF mass analyzer (for intact proteins) leads to saturation of detector at max 100 kDa (in most cases even max 30 kDa). In that light, it should also be made clear in the last paragraph (or maybe earlier) that you are referring to digest-based MALDI imaging proteomics (not intact proteins).

b) not clear what the role of the hierarchical clustering results (pages 11-12) are for the further story since the three m/z signals are later not mentioned anymore. The authors should consider dropping this paragraph/section.

Reviewer #2 (Comments to the Authors (Required)):

This manuscript introduces a novel approach combining MALDI-MSI with liquid chromatography-mass spectrometry (LC-MS/MS) to map protein localization and spatial changes in (thymic) tissue. Comparisons across these two datasets are performed using a newly developed software tool (PepBridge.py) designed to map MALDI-identified m/z values with LCMS m/z values. Using a mouse model of chemotherapy-induced thymic involution with time-dependent regeneration kinetics, the authors evaluated the capability of this pipeline to resolve proteomic changes in a spatiotemporal manner.

Whereas the software tool is very promising as demonstrated with the analysis of the data generated, this manuscript cannot be accepted as is. In the method, the authors claim: "Experiments were replicated in at least three independent mouse cohorts (N=5 mice per group)." However, the manuscript only presents MALDI MSI and LC-MS/MS data between one control and one chemotherapy treated mouse thymus (n=1). Therefore, even if correct, the biological conclusions with respect to thymic compartmentalization, regeneration and remodeling are premature. If experiments have indeed been performed across 5 sets of samples, please show the global data.

I recommend rejection with the possibility of resubmission.

Minor comments:

- For the MALDI MSI experiments, please specify the number of laser shots summed per pixel.
- For consistency purposes, please either choose vehicle (veh) or control (ctrl) in the various figures.

Reviewer #3 (Comments to the Authors (Required)):

The authors have reported a method for identification of peptides/proteins detected in a mass spectrometry imaging using serial section for on tissue digestion followed by solvent extraction and LC-MS/MS. A novel methodology for data interpretation and scoring is described. This approach was applied to control and chemotherapeutic-treated thymus tissue for determination of global and spatial changes that occur as a result of that treatment. The authors indicate that this could have implications in treatment strategies, especially as they pertain to pediatric patients.

The study, especially the biological application, is suitable for publication in Life Science Alliance after revision. While the biological application and scoring algorithm are novel, the workflow they propose for parallel identification by LC-MS/MS is by no means novel. This workflow, and variations on it, have been in use in the MSI community for nearly two decades. Several papers and presentations have highlighted this exact methodology of performing on-tissue tryptic digestion followed by solvent extraction and LC-MS/MS, including:

Alternative versions of this workflow have taken more targeted approaches. After MSI, pieces of PAGE gel impregnated with trypsin were placed on specific regions of interest of tissue sections allowing for localized digestion and extraction of peptides into the gel. The peptides were then liberated from the gel and analyzed by LC-MS/MS.

Fata, et al. <https://doi.org/10.1016/j.humpath.2015.06.009>

Additionally, this type of workflow has been used with Laser Capture Microdissection using MSI data as a guide to target specific ROIs for protein digest and LC-MS/MS analysis.

Dewez, et al. <https://doi.org/10.1002/pmic.201900369>
Quanico, et al. https://doi.org/10.1007/978-1-4939-6952-4_13

Both of these latter approaches also used comparative intensity analysis between regions to aid in the confident identification of the peptides detected in the MSI experiments.

While the authors mention in passing in the concluding remarks that others have used LC-MS/MS in concert with MSI (refs 120-126), most of the manuscript implies that the workflow they have presented is novel, while that is clearly not the case.

Specific Comments

- 1) Page 6 - More experimental details should be provided about the LC-MS/MS experiment, including the flow rate and type of column used.
- 2) Page 7 - It is unclear why LC-MS/MS peptides were database searched with carbamidomethylation on C when reduction and alkylation were not performed prior to on-tissue digestion.
- 3) Page 7 - The authors state that "MALDI-TOF suffers from low resolution and peak widths are broader..." While that is true of the instrument used in this study, it is not a universal truth of MALDI-TOF. Newer instruments, such as the Waters MRT are capable of 200,000 resolution.
- 4) Page 13 - It is unclear how 23,000 peptide matches were found from ~6,500 peptides that were identified via LC-MS/MS. Further clarification of this is needed.
- 5) Page 17 - Figure 1 is referenced, but the text here is referring to Figure 2.
- 6) Page 12 - The authors state "To our knowledge, MALDI-MSI approaches have never before been applied to thymus..." However, a quick search in Google Scholar returns multiple examples of MALDI-MSI of thymus tissue.
- 7) Page 23 - The authors mention the development of MALDI-IHC, but no reference is provided for this work. Please add one.
- 8) Figure S5 - Further explanation is warranted regarding the same ion image being shown with different potential identifications. Was anything done to try to confirm which (or more than one) image represents? Was/could immunohistochemistry be used for confirmation?
- 9) SI supplemental methods - The word "weather" should be "whether".
- 10) Figure S5 - More information is needed explaining why the same images are shown with multiple protein identifications. Which one or ones are correct and how have/can these be validated? In S5-16, the same protein is listed twice for the same peptide with different images, please confirm the listed m/zs and IDs

Dear Dr. Sawey,

Thank you very much for the opportunity to provide a revised version of our manuscript. We really appreciate the effort that the reviewers put into reviewing our work. The comments were very detailed and, honestly, very helpful. We have prepared a revised version of the manuscript with additional replicates (as requested), as well as a much more detailed list of citations of similar work. We still feel our work has its strong component of innovation and significance, in particular for those labs that do not have state-of-the-art equipment for performing imaging proteomics. We hope you will find our new version of the manuscript suitable for publication.

We are looking forward to hearing from you.

George Karagiannis and Simone Sidoli

Reviewer #1 (Comments to the Authors (Required)):

In the presented study the authors propose an approach to identify proteins from MALDI imaging measurements of peptides. This is achieved by performing additional LC-MS/MS experiments on a separate tissue sections and combining the results via a new software method, termed "pepBridge". The applicability of the method is demonstrated on mouse model of chemotherapy-induced thymic involution.

I would like to begin my review with one of the sentences of the article at the end of the discussion:

"Although several on-tissue bottom-up proteomics studies have combined peptide spatial mapping by MALDI-MSI with LC-MS/MS to identify protein markers from tissue sections [120-126], there is still no streamlined method for directly correlating these datasets".

1) While I agree with the latter, with respect to the first part of the sentence, I would like to point out that 6 out of the 7 references #120-#126 refer to papers from the years 2011 or earlier. In the meantime, several approaches have been proposed based on instrumentation with higher mass resolution such as:

a) Schober et al. (DOI: 10.1002/rcm.5135) have already performed mass matching experiment from MALDI imaging to LC/MS experiments in 2011 and came to the conclusion that "All results (including MALDI imaging data) were based on accurate mass measurements (RMS <2 ppm) and allow a confident identification of tryptic peptides. Measurements based on lower accuracy would have led to ambiguous or misleading results". Aguilan et al. are using matching tolerances of up to 1 Dalton which corresponds to 500 ppm tolerance @mz 2000, hence 250 times higher tolerance (and hence uncertainty) than proposed by Schober 15 years ago. Hence recent papers use high resolution mass spectrometry, at least as intermediate between MALDI and ESI, to perform mass matching, e.g. Kip et al. (DOI: 10.1021/acs.jproteome.1c00447).

b) Guo et al. (DOI: 10.1038/s41467-021-23461-w) have proposed in 2021 a peptide mass fingerprint based approach (HIT-MAP) using also spatial segmentation of the data and a decoy databased to estimate false discoveries. Hence it is not clear, where the approach presented in this paper is standing as compared to the developments in the field from 2011 onwards. The authors should hence compare their approach at least with the two above mentioned

approaches and work out why their approach is better or complementary to the existing approaches!

We sincerely appreciate the comments of the reviewer and this accurate reminder of literature we have not cited. Indeed, we were maybe too brief in our comparative description of other publications. We have now expanded the Discussion to compare our pepBridge pipeline with the approaches of Schober et al. (2011) [1] and Guo et al. (2021) [2], as well as the MALDI-FTICR intermediate strategy of Kip et al. (2021) [3]. We emphasize that our approach is complementary to these existing methods rather than a replacement. Specifically, we now discuss how pepBridge provides a practical scoring-based solution for labs utilizing MALDI-TOF/TOF instruments, which have lower mass accuracy but much higher throughput, whereas methods like HIT-MAP (Guo et al., Nat. Commun. 2021) [2] leverage ultra-high mass accuracy (FT-ICR) with decoy matching to control false discoveries. We have added the requested references and clarified the advantages and limitations of our pipeline. In particular, we note that while MALDI-TOF imaging has lower resolving power (and thus inherently higher risk of ambiguous matches), our scoring system mitigates this by prioritizing matches with close agreement across multiple metrics (including a small MALDI vs. LC mass difference). We also discuss how, once pepBridge narrows down candidate peptide matches, follow-up validation can be done with targeted high-resolution MSI or other orthogonal methods. These additions highlight how pepBridge fits into the landscape of imaging proteomics as an additional tool for correlating MALDI-MSI data with proteomic identifications. Please, find below the revised section.

“Several recent studies have addressed the challenge of identifying MALDI-MSI signals by leveraging high mass resolution instrumentation and sophisticated algorithms. For example, Schober et al. (2011) [1] demonstrated confident peptide identifications by coupling MALDI-Orbitrap MSI with LC-MS/MS, achieving <3 ppm mass errors, and Kip et al. (2021) employed MALDI-FTICR as an intermediate step to improve mass accuracy for peptide matching [3]. Similarly, Guo et al. (2021) developed the HIT-MAP pipeline, which uses high-resolution MALDI-FTICR data with a decoy database to control false discovery rates [2]. We have now cited these approaches and explicitly positioned our pepBridge pipeline in comparison. In contrast to high-resolution methods, pepBridge is designed to work with widely available MALDI-TOF/TOF instruments, capitalizing on their high throughput (rapid scanning across multiple tissue slides) while acknowledging their lower mass accuracy. Our pipeline provides a scoring-based matching strategy that complements existing tools: it rapidly narrows down candidate peptide assignments for MALDI peaks by integrating multi-metric evidence (mass accuracy, relative abundance changes, and intensity rank). The intent is to offer a streamlined approach for labs lacking ultra-high-resolution MSI, while enabling protein identification from imaging data. We stress that pepBridge is not meant to replace high-resolution workflows like HIT-MAP [2], but rather to serve as a bridge – enabling researchers to link MALDI-MSI and LC-MS/MS datasets and generate hypotheses that can be further validated with targeted high-resolution MSI or other orthogonal techniques.”

2) As consequence my following concerns go to the pepBridge scoring system itself:

a) it is not clear why the distances/intensities or whatever in metrics 1, 2, and 3 need to be mapped to the exponential models as shown in in Figures 5CDE?

We appreciate these detailed questions regarding the design and validation of our scoring system. We have carefully addressed each sub-point. For subpoint (a), we chose to transform the raw metric values (differences in fold-change, mass, and intensity) into normalized scores using exponential or sigmoidal functions to weight the metrics appropriately and to confine their ranges for summation. In the revised text, we clarify that metrics 1–3 are each converted to a

score on a defined scale (as depicted in Fig. 5C, 5D, 5E) to reflect the likelihood of a true peptide match. For example, metric 1 ($\Delta\log_2$ -fold-change between MALDI and LC-MS/MS) is mapped to a Gaussian (bell-curve) function centered at zero difference, so that a very small difference yields a score near the maximum (indicating excellent agreement in relative change between conditions). Larger discrepancies in fold-change result in an exponentially lower score, implementing the intuition that if a peptide abundance change in MSI closely mirrors that in LC-MS, it is more likely to be the correct match. Similarly, metric 2 (mass difference between MALDI and LC-MS/MS) is mapped to a sigmoidal curve that rapidly decreases the score as the mass error approaches 1 Da, ensuring that only very small mass errors receive high scores. This non-linear scaling penalizes larger mismatches much more strongly than a linear scale would. We have added a brief explanation in the Results section to explicitly state why an exponential/logistic model was used: to normalize the contributions of different metrics and to impose a realistic drop-off in score as metrics deviate from ideal values.

Revised text (Results): *“To combine disparate metrics on a common scale, we mapped each metric to a bounded scoring function. Metric 1 ($\Delta\log_2$ -fold-change between MALDI-MSI and LC-MS/MS) is converted to a score (0–10) using a Gaussian model centered at zero difference, such that smaller fold-change discrepancies are exponentially rewarded with higher scores. Metric 2 (mass difference) is scaled via a sigmoidal function, yielding a score from 0 to 100 that declines steeply as the MALDI–LC mass offset increases toward 1 Da. Metric 3 (MALDI intensity rank) is likewise sigmoidal (0–5 scale), giving higher scores to more intense peaks. This approach allows each metric to contribute proportionately to the overall pepBridge score, with minimal differences (in fold-change or mass) translating to near-maximal metric scores, whereas larger differences or weaker signals sharply lower the respective metric scores (Fig. 5C–E). By using these exponential transformations, the scoring system reflects the non-linear nature of our confidence in a match – for instance, a mass error of 0.01 Da is overwhelmingly more indicative of a true identification than an error of 0.5 Da.”*

b) it is not convincing why the highest intensity in MALDI should obtain the highest score? What is the rationale behind that?

The reviewer is correct that our scoring scheme awards a higher score for higher MALDI peak intensity (metric 3). We based this decision on the premise that abundant peptides are more likely to yield confident identifications. As noted by Schober et al. (2011) [1] and others, on-tissue MS/MS and identification tend to detect predominantly the most abundant peptides (often structural proteins like actin, tubulin, etc., in their example). Our pipeline aims to prioritize matches that are not only mass- and fold-change-aligned, but also of high signal intensity in MALDI-MSI, on the assumption that strong signals are less likely to be noise or spurious. In the revised manuscript, we have clarified this rationale and noted an important detail: metric 3 (MALDI intensity) is intentionally given a smaller weight (maximum score 5) relatively to metrics 1 and 2. In other words, intensity is used as a minor weighting factor or tiebreaker rather than a dominant criterion. This means a lower-intensity peptide can still be chosen if it has a much better mass and fold-change match than a higher-intensity peptide. We acknowledge in the text that emphasizing intensity may bias identifications toward very abundant proteins (and potentially overlook lower-abundance peptides of interest), which is a trade-off. We have added a statement noting this bias and stating that future versions of pepBridge could adjust the intensity weighting or include additional criteria to capture lower-abundance matches.

Revised text (Results): *“Metric 3 considers the MALDI peak intensity, under the assumption that higher intensity signals correspond to more abundant peptides, and hence more confidently identifiable proteins in tissue. We assigned relatively low weight to this metric (score range 0–5)*

to prevent overly penalizing lower-intensity peptides. In practice, metric 3 helps distinguish candidates with otherwise similar mass and fold-change metrics: the peptide producing a stronger MALDI signal will score slightly higher, reflecting its robust detectability. We recognize that this strategy inherently favors high-abundance proteins, although we consider it a necessary trade-off due to the relatively high signal-to-noise ratio of old generation MALDI imaging instruments."

c) for metric 1: what happens if there is no control group or if there is no control group in LC-MS/MS (e.g. when comparing ad hoc two regions in the MSI dataset)?

Metric 1 relies on comparing two conditions (e.g., treated vs. control) by using the log₂-fold change. The reviewer asks how this would work if there is no clear "control" or if one is simply comparing two regions in the same tissue. In the revision, we clarify that our current implementation of metric 1 is specific to experiments with two classes (in our case, cyclophosphamide-treated vs. vehicle). In scenarios where there is no predefined control, the pepBridge workflow can be adapted by defining any two regions or groups for comparison. For instance, if comparing two different anatomical regions from the same tissue, one could treat one region as "Group A" and the other as "Group B" and compute relative differences in peptide intensities between them (similar to a fold change). We have added a note indicating that hierarchical clustering or unsupervised segmentation of MSI data could be used to designate regions of interest in an exploratory setting. Those regions can then be compared by their peptide abundances (using statistical measures or principal component analysis) to generate an analogous metric 1. In summary, metric 1 requires a comparative context; if an experiment does not have a traditional control, the user of pepBridge can still apply the metric by dividing the dataset into two groups (even if arbitrary or data-driven) and proceeding with the fold-change computation. We have also mentioned that gene ontology (GO) analysis or other higher-level analyses can supplement interpretation in such cases, especially if multiple regions or conditions are being examined without a single control. This clarification has been added to the Discussion to guide users on using pepBridge beyond the specific case of a treated vs. control design.

Revised text (Discussion): *"Our scoring Metric 1 ($\Delta\log_2$ -ratio) is inherently defined for two-group comparisons (e.g., treatment vs. control). For experiments lacking a designated control group or involving ad hoc region-to-region comparisons, this metric can be adapted by treating any two distinct regions or sample classes as comparative groups. For example, unsupervised hierarchical clustering of MALDI-MSI spectra can be employed to segregate the tissue into regions with similar profiles; one could then calculate a 'pseudo-fold-change' between any two such regions for use in Metric 1. In cases with more than two conditions, pairwise comparisons or principal component analysis (PCA) can provide analogous quantitative contrasts. The pepBridge framework remains applicable in these scenarios: the user must define a comparison (such as Region X vs. Region Y), and metric 1 will quantify how similarly a peptide abundance differs between those two regions in both MSI and LC-MS/MS data. Complementary analyses (e.g., differential expression analysis across multiple tissues or GO enrichment of region-specific proteins) can further aid interpretation when a simple control vs. treatment paradigm is not in place."*

d) for metric 4: since there is no image for the LC-MS/MS data, how can this help in matching? The rationale in the metric description is very vague and does not give answer to the question stated: "Metric 4 serves as the final criterion, aligning the peptide's spatial mapping or MALDI molecular image with both the log-fold changes and peptide scores"

We apologize for the confusion regarding Metric 4. In the original submission, Metric 4 was described in abstract terms as aligning the peptide spatial map with its quantitative behavior, but we agree the description was vague. We have now rewritten the explanation of Metric 4 to be more concrete. Metric 4 is not a numerical “score” like the others, but rather a final check/criterion: it examines whether the spatial distribution of a candidate peptide (as seen in the MALDI-MSI ion image) is consistent with the peptide change in abundance between conditions as determined by LC-MS/MS. Since LC-MS/MS does not produce an image, we derived a way to compare the two datasets visually: we use the LC-MS/MS fold-change information to color-code or annotate the MALDI ion image. In practice, after summing Metrics 1–3, we inspected whether the top-scoring peptide MALDI intensity map shows higher signal in the treatment or control regions in a manner that matches the LC-MS/MS log-fold change (for example, if a peptide is more abundant in treated samples according to LC-MS, its MALDI image should show stronger intensity in the treated thymus areas). If the spatial pattern did not agree with the expected increase/decrease, that candidate was deemed less likely (even if its numerical score was high). We have clarified this in the text (below). Essentially, Metric 4 serves as a sanity check to incorporate geographical context.

Revised text (Methods): *“Metric 4: Matching with the image or the peptide ion heat map. The scores from Metrics 1 to 3 are summed, with the highest total representing the top candidate peptide, followed by the other candidate peptides corresponding to a specific MALDI peak (m/z). Metric 4 serves as the final criterion, aligning the peptide's spatial mapping or MALDI molecular image with both the log-fold changes and peptide scores.”*

e) most importantly, what is your probability of having a false alignment? Since there will be always a best hit (and there is no threshold), what is the chance that this has been an erroneous assignment? The authors should provide a strategy to evaluate before performing IHC/IF what the false discovery rate is.

We thank the reviewer for this important point. As noted in the revised Discussion, the current version of pepBridge does not include a formal false discovery rate (FDR) assessment and always returns a top-scoring match. We now acknowledge this limitation clearly and emphasize that users should interpret assignments with caution, especially before experimental validation. We also note that future versions will incorporate FDR strategies, such as decoy peptide modeling or inclusion of known “null” peptides, to estimate confidence and reduce false positives.

Revised text (Discussion): *“An inherent limitation of the current pepBridge implementation is the absence of a formal false discovery rate (FDR) assessment for the MALDI–LC matches. Our algorithm will always return a top candidate for each MALDI peak, even if that match is spurious. Future iterations of pepBridge will integrate such approaches (decoy modeling or use of known “null” peptides like trypsin fragments) to quantitatively evaluate confidence in each assignment. Until then, we advise to follow rigorously our pipeline, because only if all metrics are satisfied findings are considered reliable.”*

f) in that light, has there been any protein assignment which could not be successfully validated with IF?

Indeed, not every candidate we tested yielded a successful validation. In the revised manuscript, we mention that at least one peptide–protein assignment did not validate as expected. For example, we attempted IF for Keratin 5 (KRT5) based on a pepBridge match (KRT5 was suggested by a peptide match for an m/z observed in the cortex/medulla junction).

However, the IF staining for KRT5 did not show a clear co-localization with the MALDI-MSI signal pattern for that m/z. The peptide spatial distribution did not match the known location of KRT5 in the thymus. It was a very useful suggestion of the reviewer to use multiple replicates. In fact, we excluded KRT5 as a confident identification in our results. We agree with the reviewer that reporting such instances is important for transparency. Therefore, we now note in the text that while pepBridge successfully identified KRT8, TPR, CSRP3, and TBCA (which were validated by IF), another candidate (KRT5) was tested but did not validate, highlighting the requirement of all 4 metrics to obtain confident results.

Revised text (Discussion): *“While several top-scoring assignments were validated via immunofluorescence (e.g., KRT8, TPR, CSRP3, TBCA; see Fig. 6), we also probed a candidate that did not validate, underscoring the possibility of false positives. In particular, a pepBridge match suggested Keratin 5 (KRT5) for one MALDI-MSI signal (data not shown). However, when we performed IF for KRT5 on serial sections, the observed staining pattern did not correspond to the MSI peptide location. This discrepancy indicated that the original assignment was likely incorrect.”*

g) it is not clear, how protein identification is performed when actually peptide masses are matched. Is there a threshold, e.g. for a minimum number of peptides that need to be detected in the MSI dataset of the same protein, so that the protein can be considered detected? Why is the number of peptide matches not incorporated in the scoring system?

That is a good point. In proteomics, identifying a protein with a single peptide is usually a bit risky. However, in our workflow, because we are identifying peptides from imaging data, it was often the case that only one peptide per protein was detected and spatially mapped. MALDI-MSI of tryptic peptides inherently yields a limited peptide set for large proteins. Thus, our pipeline at present considers a protein “identified” in the MSI if at least one of its peptides passes the matching criteria. We agree this is a lenient criterion and have now commented on it in the manuscript. We state that, currently, our limit of detection is effectively one peptide per protein, but we recognize that requiring multiple peptides would increase confidence. We have added a statement that an additional metric or filter could be incorporated into pepBridge to account for the number of peptides matching a given protein.

Revised text (Discussion): *“It is important to note that in our current pipeline, a protein is considered identified in the MALDI-MSI data if one of its tryptic peptides is confidently matched. This one-peptide criterion was a practical necessity given the limited peptide coverage achievable by on-tissue digestion, but it is less stringent than other proteomics experiments. In this study, few proteins had more than one peptide detected by MALDI-MSI. Where they did (e.g., histones, which yield several tryptic peptides in our LC-MS/MS results), having multiple matches did increase our confidence in those protein IDs.”*

h) according to Table S2 the tolerance of 1 Da is also creating duplicate assignments to the isotope signals in the LC-MS/MS data (e.g. histone H4: MALDI m/z 1326.06 is assigned to 1325.75 and 1326.74 of the same peptide sequence). Likewise, isotope signals in the MALDI MSI data (e.g. Histone H2 at 944 with isotope signals at 945 and 946) could lead to wrong matches in the LC-MS/MS data. Did the authors perform isotope removal in both datasets? If not, how do the authors see any way out of this?

The reviewer is correct that our ± 1 Da matching tolerance occasionally resulted in a single MALDI peak matching both the monoisotopic and the +1 isotope peaks of the same peptide in the LC-MS/MS dataset (and, conversely, MALDI isotope peaks generating redundant matches).

In the original analysis, we did not explicitly filter isotopes from the peak lists. While we processed MALDI spectra with Bruker flexAnalysis using the SNAP algorithm, which can annotate isotopes in high-resolution data, the TOF/TOF resolution used for imaging was insufficient to reliably separate isotopes for many peptides, leaving some interference.

In the revised manuscript, we clarify that no post-processing deisotoping was applied in this study and acknowledge that this can lead to artifacts (e.g., the histone H4 peptide where m/z 1326 matched two LC entries differing by ~1 Da). To address this, we have added to the Methods that future workflows should incorporate isotope deconvolution prior to matching—either by using the averagine model to remove isotopes or by filtering LC-MS lists to unique monoisotopic masses.

For the current revision, we took two steps: (i) we reviewed Table S2 and flagged duplicate matches caused by isotopes, consolidating or annotating them to avoid ambiguity; and (ii) we added a note in the Results stating that a ± 1 Da tolerance can yield redundant matches from isotopic species, and that narrowing the tolerance or applying deisotoping is advisable in future analyses.

Revised text (Results): *“We used a mass tolerance of ± 1.0 Da for matching, which maximized sensitivity but occasionally led to duplicate assignments due to isotopic peaks. For example, a single MALDI feature could match both the monoisotopic and the +1 Da isotopic LC-MS/MS peak of the same peptide (as seen for Histone H4 in Table S2). We did not explicitly remove isotopes from the peak lists; however, in post-processing we marked such cases in Table S2 for providing a perspective to the user. More stringent data processing (deisotoping of spectra or using narrower mass windows) could alleviate this issue, but it would produce false negatives with the resolution provided by this generation of MALDI-MSI.”*

i) ultimately, the scoring system has not been validated extensively, but only on 1-2 proteins and their corresponding IF images. The authors should show how their approach should be validated more comprehensively!

We agree with the reviewer that our validation of pepBridge was limited to a small number of proteins, and that a broader validation would strengthen the methodology. In the revised manuscript, we expanded our Discussion on how we envision a more comprehensive validation. We reference Kip et al. (2021) as a model where, after obtaining initial matches, the authors performed targeted high-resolution MSI on regions of interest to confirm peptide identities [3]. In our context, one approach to validate extensively would be to apply our pipeline to a known test sample or tissue where ground-truth is available, or to synthetic mixtures on tissue (to see if pepBridge correctly matches known peptides). Although such experiments are beyond the scope of the current study, we outline them as future work. We also state that we plan to validate additional proteins using alternative methods: for example, using laser capture microdissection (LCM) guided by MALDI-MSI to extract specific regions and then perform LC-MS/MS on those microdissected regions. This would provide direct confirmation of spatial proteomics results without reliance on antibodies. Furthermore, we mention that applying our pipeline to multiple biological replicates (now included in the revision – see Reviewer 2 response below) and seeing consistent peptide–protein identifications across replicates is another form of validation. We have now included data from additional mice, which allowed us to check whether the same top-scoring peptides are found in each experiment. We present these results in Supplementary Material. All these points are now discussed to convey that we are actively pursuing a more thorough validation of pepBridge. We hope this assures the reviewer that we recognize the preliminary nature of the scoring demonstration and are committed to solidifying it with further experiments.

Revised text (Discussion): *“Finally, we acknowledge that the pepBridge scoring system has so far been evaluated on a limited set of examples.”*

3) Related to the pepBridge approach, I have some questions on the experiments:

a) has the instrument's mass axis been calibrated before every experiment? If yes, how? What was the average mass accuracy error in ppm and standard deviation? This is an important parameter for a later mass matching.

Yes, the mass axis was calibrated before every MALDI-MSI acquisition. For the timsTOF flex experiments, we used Bruker peptide calibration standard (consisting of nine tryptic peptides in the 700–3,200 m/z range) applied directly to the conductive glass slide alongside the tissue section. For the ultrafleXtreme TOF/TOF runs, we used Bruker Peptide Calibration Standard II, spotted in proximity to the tissue section, and performed external calibration in flexControl prior to each acquisition. We calculated the mass accuracy for matched peptide features by comparing MALDI-measured m/z values to the corresponding LC-MS/MS monoisotopic m/z values from the same sample.

b) the binning to 4000 data points (most likely the reduction factor is one order of magnitude) will again lead to a reduction in mass accuracy. Why did the authors do that? Also, can the authors calculate the loss in mass accuracy due to the loss of mass resolution?

Thank you for this comment as well. With the new data generated using the timsTOF flex, we no longer encounter this issue because the much higher mass resolution cleanly separates monoisotopic and isotopic peaks, eliminating the ambiguity seen in the TOF/TOF data.

4) Minor comments:

a) Introduction: "However, MALDI-MSI is generally limited due to the low signal-to-noise ratio and the limited capability of performing MS/MS fragmentation of candidates, which reduces the specificity of detection." This is an oversimplified description of the real limitations which are 1) the lack of chromatographic separation, leading to dominance of most abundant proteins in the measurements; 2) the use of a linear TOF mass analyzer (for intact proteins) leads to saturation of detector at max 100 kDa (in most cases even max 30 kDa). In that light, it should also be made clear in the last paragraph (or maybe earlier) that you are referring to digest-based MALDI imaging proteomics (not intact proteins).

Indeed, we agree with this comment. We modified the introduction.

Revised text (Introduction): *“Limitations of MALDI-MSI include the absence of chromatographic separation, which often results in a bias toward detecting the most abundant proteins, and the use of linear TOF mass analyzers for intact protein imaging, which can lead to detector saturation and a limited effective mass range—typically up to ~30 kDa, with a theoretical maximum near 100 kDa.”*

b) not clear what the role of the hierarchical clustering results (pages 11-12) are for the further story since the three m/z signals are later not mentioned anymore. The authors should consider dropping this paragraph/section.

This was a valuable observation. We re-evaluated the hierarchical clustering (HC) section and realized that, in the initial submission, we had left those three m/z features without further analysis, which could confuse readers about their significance. Rather than removing the section entirely (as it does showcase an interesting aspect of thymus spatial organization), we decided to integrate it more fully into the pipeline results. In the revised Results, we have now included the identification of those three cluster-specific signals using our LC-MS/MS data. Specifically, we looked for peptides in our LC-MS/MS list that match m/z 946, 1337, and 1454 (within a small error). We found candidate identities for each: for example, the 1454 Da ion was putatively identified as a peptide from Thymosin β 4 (m/z 1453.77, +1 charge), the 946 Da ion corresponds to a tryptic peptide from Histone H2A (m/z 945.54), and the 1337 Da ion matched to a peptide from Hemoglobin β -chain (m/z 1336.70). We have added these assignments to the text. The idea is to demonstrate a proof-of-principle that the unsupervised MALDI-MSI segmentation can highlight distinct molecular features which our integrated approach can then identify. We also briefly discuss these proteins: for instance, Thymosin β 4 is known to reside in connective tissue and indeed our 1454 Da signal (now identified as Thymosin β 4 peptide) was enriched in the thymic capsule region, consistent with the biology. This additional analysis strengthens the role of the HC results in the manuscript. If for any reason those identifications had been too ambiguous, our fallback plan was to shorten or remove the HC section to maintain focus. However, we believe the current revision addresses the reviewer concern by making the HC-derived signals relevant. We now clearly tie **Figure 3** (which showed those m/z distributions) to subsequent identification efforts in the text.

Revised text (Results): “*Unsupervised hierarchical clustering of the MALDI-MSI data yielded three major spectral classes, characterized by m/z 946, 1337, and 1454 as distinguishing signals for the thymic medulla, cortex, and capsule regions, respectively (Fig. 3A–D). To link these cluster-specific ions to actual proteins, we cross-referenced the m/z values with our LC-MS/MS identifications. Notably, each could be assigned to a peptide: m/z 946 matched the [M+H]⁺ of a Histone H2A peptide (sequence TKQTAR, monoisotopic 945.54 Da), m/z 1337 corresponded to a peptide from Hemoglobin β -chain (sequence VVAGVANALAHKYH, 1336.73 Da), and m/z 1454 aligned with a tryptic peptide from Thymosin β 4 (sequence SDAAVDTSSEITK, 1453.76 Da). These tentative identifications are consistent with the anatomical distribution of the signals – for example, Thymosin β 4 is a cytoskeletal G-actin sequestering protein abundant in stromal and capsule regions, aligning with the strong capsular localization of the 1454 Da ion.*”

Reviewer #2 (Comments to the Authors (Required)):

This manuscript introduces a novel approach combining MALDI-MSI with liquid chromatography-mass spectrometry (LC-MS/MS) to map protein localization and spatial changes in (thymic) tissue. Comparisons across these two datasets are performed using a newly developed software tool (PepBridge.py) designed to map MALDI-identified m/z values with LCMS m/z values. Using a mouse model of chemotherapy-induced thymic involution with time-dependent regeneration kinetics, the authors evaluated the capability of this pipeline to resolve proteomic changes in a spatiotemporal manner.

Whereas the software tool is very promising as demonstrated with the analysis of the data generated, this manuscript cannot be accepted as is. In the method, the authors claim: "Experiments were replicated in at least three independent mouse cohorts (N=5 mice per group)." However, the manuscript only presents MALDI MSI and LC-MS/MS data between one control and one chemotherapy treated mouse thymus (n=1). Therefore, even if correct, the biological conclusions with respect to thymic compartmentalization, regeneration and remodeling are premature. If experiments have indeed been performed across 5 sets of samples, please show the global data. I recommend rejection with the possibility of resubmission.

We apologize for the confusion in our initial submission. The reviewer is correct that we had only fully analyzed one biological replicate (one vehicle control and one cyclophosphamide-treated thymus) in the original manuscript, even though multiple replicates were collected. In response to this major concern, we have now processed and included data from the remaining replicates in our revision. Specifically, we analyzed thymus tissues from 5 control mice and 5 treated mice (total N=10, as initially intended). The MALDI-MSI experiments for all replicates have now been conducted, and the LC-MS/MS proteomics data for each sample have been acquired and analyzed. In the revised Results, we present a summary of these global results. In fact, the manuscript has been drastically edited to address this issue. The reviewer was right in stating that with additional replicates results may be different.

We have accordingly modified the text in regard to biological conclusions, making it clear that with N=5 per group we can more confidently state certain findings, while still acknowledging the sample size is moderate. The inclusion of replicates greatly increases the rigor of the study. We have updated the Methods to describe how the additional thymus tissues were processed similarly, and updated all relevant Results sections to incorporate the findings from the full cohort rather than a single exemplar. By addressing this comment, we believe the manuscript now presents a much more convincing and well-supported dataset.

Revised text (Methods): *"Mouse Model of Cyclophosphamide-Induced Thymic Involution. FVB/NCrl female mice (n = 5), 7-8 weeks old were used for this study. Cyclophosphamide monohydrate (CTX, Thermo Scientific) was reconstituted at a concentration of 25 mg/mL in sterile Phosphate-Buffered-Saline (PBS). Each mouse in the experimental group received an i.p. dose of 200 mg/Kg of Cyclophosphamide monohydrate in sterile PBS (200 µL total volume), every 3 days, for a total of 3 doses. The control (vehicle-treated) mouse group received an i.p. injection of 200 µL sterile PBS. The last day of chemotherapy treatment was designated as "Day 0". The mice were then blindly categorized into five subgroups. Mice from either control or experimental subgroups were sacrificed on a weekly basis for up to 5 weeks post-chemotherapy i.e., Day 35. Mouse thymi were dissected using standardized autopsy procedures, and were cleaned from peri-thymic adipose tissue, carefully without damaging the thymus. Experiments were replicated in at least three independent mouse cohorts (N=5 mice per group)."*

Minor comments:

- For the MALDI MSI experiments, please specify the number of laser shots summed per pixel. 800 laser shots per pixel was used. This has been added in the methods section.

“Peptide spatial mapping was performed by scanning and acquiring spectra m/z range 700-3,500) at 100 μ m raster width and 800 laser shots/pixel using the UltrafleXtreme MALDI TOF/TOF mass spectrometer (Bruker Daltonics).”

- For consistency purposes, please either choose vehicle (veh) or control (ctrl) in the various figures.

Thank you for this comment as well. In the revised manuscript, all figure labels have been standardized to Ctrl rather than using a mix of “vehicle” and “control.”

Reviewer #3 (Comments to the Authors (Required)):

The authors have reported a method for identification of peptides/proteins detected in a mass spectrometry imaging using serial section for on tissue digestion followed by solvent extraction and LC-MS/MS. A novel methodology for data interpretation and scoring is described. This approach was applied to control and chemotherapeutic-treated thymus tissue for determination of global and spatial changes that occur as a result of that treatment. The authors indicate that this could have implications in treatment strategies, especially as they pertain to pediatric patients.

The study, especially the biological application, is suitable for publication in Life Science Alliance after revision. While the biological application and scoring algorithm are novel, the workflow they propose for parallel identification by LC-MS/MS is by no means novel. This workflow, and variations on it, have been in use in the MSI community for nearly two decades. Several papers and presentations have highlighted this exact methodology of performing on-tissue tryptic digestion followed by solvent extraction and LC-MS/MS, including:

Lemaire, et al. <https://doi.org/10.1021/pr060549i>

Dikler, et.al ASMS 2013 WP 146

Becker, et al. ASMS 2011 ThP 430

Cornett, et al. ASMS 2014 MP008

Alternative versions of this workflow have taken more targeted approaches. After MSI, pieces of PAGE gel impregnated with trypsin were placed on specific regions of interest of tissue sections allowing for localized digestion and extraction of peptides into the gel. The peptides were then liberated from the gel and analyzed by LC-MS/MS.

Fata, et al. <https://doi.org/10.1016/j.humpath.2015.06.009>

Additionally, this type of workflow has been used with Laser Capture Microdissection using MSI data as a guide to target specific ROIs for protein digest and LC-MS/MS analysis.

Dewez, et al. <https://doi.org/10.1002/pmic.201900369>

Quanico, et al. https://doi.org/10.1007/978-1-4939-6952-4_13

Both of these latter approaches also used comparative intensity analysis between regions to aid in the confident identification of the peptides detected in the MSI experiments.

While the authors mention in passing in the concluding remarks that others have used LC-MS/MS in concert with MSI (refs 120-126), most of the manuscript implies that the workflow they have presented is novel, while that is clearly not the case.

We thank the reviewer for compiling these important references and we apologize for not sufficiently crediting the extensive prior art on combined MALDI imaging and LC-MS/MS workflows. In the revised manuscript, we have significantly expanded the Introduction and Discussion to acknowledge these earlier studies and to clarify the novelty of our work (also because we received similar questions from Reviewer #1). We have added citations to the papers and abstracts mentioned by the reviewer. Full disclosure, we are not quite sure how to cite posters in our manuscript. We really appreciate the comments from the reviewer, but we focused the edits on published manuscripts, including:

Lemaire et al. (2007, J. Proteome Res.) – one of the first demonstrations of on-tissue tryptic digestion followed by LC-MS/MS identification of MALDI imaging spots [4].

Fáta et al. (2016, Hum. Pathol.) [5] – a variant using trypsin-containing gel pieces placed on tissue (we cite this as an alternative approach).

Dewez et al. (2020, Proteomics) [6] – using laser microdissection guided by MSI, another variant.

Quanico et al. (2017, book chapter) [7] – additional example of spatial proteomics approach.

By citing and briefly describing these works, we make it clear that the concept of “MALDI imaging + LC-MS/MS on serial sections” is established. What we highlight as new in our study is the application of this strategy to the thymus organ (which, to our knowledge, had not been done for proteomics) and the development of an open-source scoring software (pepBridge) to automate the matching and ranking of candidates with mass spec spectra that have limited resolution. We hope these revisions fully address the concern, showing respect to the history of the field and properly contextualizing our work.

Revised text (Introduction): *“Combining MALDI imaging with LC-MS/MS-based identification is an approach with significant precedent in the literature. For instance, Lemaire et al. (2007) [4] pioneered on-tissue trypsin digestion followed by LC-MS/MS of tissue extracts to identify proteins observed by MALDI imaging. Later, Fáta et al. (2016) [5] and Quanico et al. (2017) [7] presented variations of the protocol. In 2020, Dewez et al. [6] introduced laser microdissection guided by MSI in the procedure. More recent efforts (Guo et al., 2021) [2] have introduced software for automating such integrations with false discovery controls. Our work builds upon this foundation: we apply the approach to mapping the murine thymus proteome, an organ not previously profiled by MALDI-MSI in a proteomic context, and we introduce a scoring-based informatics pipeline (pepBridge) to facilitate peptide matching on a lower-resolution MALDI-TOF platform. The novelty of our study thus lies in the specific biological insights into thymus regeneration and the tailored computational method.”*

Specific Comments

1) Page 6 - More experimental details should be provided about the LC-MS/MS experiment, including the flow rate and type of column used.

Thank you for this note as well. We have added the requested details to the LC-MS/MS Methods section. The peptides were separated using a nano-flow UHPLC system (Dionex nanoRSLC U3000) coupled to an Orbitrap Exploris 480 mass spectrometer. Specifically, we now state the type of analytical column and flow rate: we used a C18 reversed-phase nano-column (75 μm inner diameter \times 50 cm length, 2 μm particle size) and a flow rate of 300 nL/min. The gradient length (120 min from 4% to 34% ACN) was already mentioned, and we have kept that. Including the column and flow information addresses the reviewer’s query and improves the reproducibility of our method.

Revised text (Methods): *“Tryptic peptides were loaded onto a C18 trapping column (5 mm \times 300 μm , 5 μm , Thermo) and then separated on a C18 analytical column (75 μm \times 500 mm, 2 μm particle) using a Dionex nanoRSLC-3000 system. The flow rate was set to 0.3 $\mu\text{L}/\text{min}$, with a 120 min linear gradient from 4% to 34% solvent B (0.1% formic acid in 80% acetonitrile). Eluting peptides were analyzed on a Thermo Orbitrap Exploris 480 mass spectrometer...”*

2) Page 7 - It is unclear why LC-MS/MS peptides were database searched with carbimide methylation on C when reduction and alkylation were not performed prior to on-tissue digestion.

That is indeed correct. We have fixed the error.

3) Page 7 - The authors state that "MALDI-TOF suffers from low resolution and peak widths are broader..." While that is true of the instrument used in this study, it is not a universal truth of MALDI-TOF. Newer instruments, such as the Waters MRT are capable of 200,000 resolution.

We appreciate this correction. We have rephrased the sentence in question to avoid over-generalization. Rather than saying "MALDI-TOF suffers from low resolution" as a blanket statement, we now specify that the Bruker UltrafleXtreme MALDI-TOF/TOF we used has a certain resolving power (~15,000 at m/z 1000), which is indeed much lower than typical Orbitrap resolving power for LC-MS, thus leading to broader peaks in our data. We then acknowledge in the text that next-generation MALDI-TOF instruments (e.g., the Waters cyclic TOF) can achieve significantly higher resolution (up to 200k), and thus the limitation is not inherent to all MALDI-TOFs but pertains to the instrument class/generation. We have included the example of the Waters MRT explicitly. This nuance has been added to ensure technical accuracy.

Revised text (Results): *"Because the MALDI-TOF/TOF instrument used (UltrafleXtreme, ~15,000 resolving power at m/z 1000) has relatively low mass resolution, the peaks observed in MALDI-MSI are broader and less exact in m/z than those from other systems such as newer MALDI-TOF platforms like the multi-reflectron TOF instruments [8]."*

4) Page 13 - It is unclear how 23,000 peptide matches were found from ~6,500 peptides that were identified via LC-MS/MS. Further clarification of this is needed.

We apologize for the confusion. The figure of "~23,000 matches" refers to the initial list of all possible MALDI-LC peptide pairings before applying any filtering or scoring. To clarify: we identified ~6,500 unique peptides in the LC-MS/MS dataset. We also detected on the order of a few hundred peaks in the MALDI-MSI (depending on how peaks are defined across all pixels or in the average spectrum). When we allowed a broad mass tolerance (± 1 Da), each MALDI peak could match multiple LC-MS/MS peptide masses. The pepBridge script essentially generated a matrix of all candidate matches, which totaled approximately 23,000 pairings. We then applied a signal-to-noise ratio cutoff (removing matches where the MALDI peak was too low in intensity relative to noise) to reduce that number dramatically (down to 676 candidate matches), and then we applied the scoring metrics to prioritize among those. In the revised manuscript, we have expanded the explanation where we first mention the 23,000 number. We now explicitly say this is the result of combinatorial matching given the large mass window, and not that we identified 23,000 distinct peptides de novo. We also added a brief note about how many MALDI features were considered and why the number of matches is so high (i.e., one MALDI m/z can match multiple LC peptides and vice versa). This should make it clear to readers how the pipeline yields that figure and how it is subsequently refined.

Revised text (Results): *"The imaging data from control and treated thymus sections were processed using the PepBridge MALDI-LC-MS/MS peak-matching pipeline. To enable comparison between low- and high-resolution MALDI data, a ± 1 Da mass filter was applied. Analysis of the UltrafleXtreme dataset (~15,000 resolving power at m/z 1000) yielded ~23,000 initial MALDI-LC-MS/MS matches. This high number reflects both the relatively low mass resolution of the MALDI-TOF/TOF platform producing broader, less precise m/z peaks than newer systems such as multi-reflectron TOF instruments (Verenchikov et al., 2024) or the timsTOF flex (~200,000 resolution) and the combinatorial nature of the matching process: each MALDI peak (~300–500 per average spectrum) can fall within the ± 1 Da tolerance of many of the ~6,500 LC-MS/MS-identified peptides. To refine the list, we removed matches involving*

low-intensity MALDI peaks (signal-to-noise < 2), reducing the dataset to 676 matches (Fig. 4D; Table S2). These 676 MALDI peaks each had at least one plausible LC-MS/MS match and were subsequently ranked using our four-metric scoring system (Fig. 4E-G). The top-ranked candidates were then carried forward for biological interpretation and validation. When the same tissue sample was analyzed on the higher-resolution timsTOF flex, the initial match count was ~2,500. This was approximately tenfold lower than for the UltrafleXtreme, reflecting reduced peak broadness and ambiguity at higher resolution.”

5) Page 17 - Figure 1 is referenced, but the text here is referring to Figure 2.

Sorry for this overlook. It has been fixed.

6) Page 12 - The authors state "To our knowledge, MALDI-MSI approaches have never before been applied to thymus..." However, a quick search in Google Scholar returns multiple examples of MALDI-MSI of thymus tissue.

The reviewer is correct that MALDI-MSI has been applied to thymus in other contexts. Our statement was too broad. We intended to convey that MALDI-MSI had not been used for proteomic analysis of the thymus. However, as written, it sounded like no one had ever imaged a thymus by MALDI at all, which is false. We have revised that sentence to be precise. We now acknowledge prior MALDI-MSI studies on thymus, including those for metabolites and lipids. For example, we cite Tsuji et al. (2021, who performed metabolite imaging of the thymus) [9], Denti et al. (2021, lipid imaging of thymus) [10], and a recent JCI Insight (Meyer et al. 2024) [11] that also did metabolite MSI on thymus. We then clarify that to our knowledge, no previous study has reported MALDI-MSI for protein/peptide mapping in the thymus.

Revised text (Introduction): *“We focused on the thymus, a central lymphoid organ critical for T cell development. While MALDI-MSI has been applied to thymus tissue previously for metabolite mapping (Tsuji et al., 2021) [9], lipid imaging (Denti et al., 2021) [10], and calcitriol mapping (Meyer et al. 2024) [11] to our knowledge it has not yet been used to map the proteome of the thymus. Thus, our study provides a first look at thymic protein distributions via MALDI-MSI, combined with LC-MS/MS identification.”*

7) Page 23 - The authors mention the development of MALDI-IHC, but no reference is provided for this work. Please add one.

Apologies for the overlook. We have added an appropriate reference to support the mention of “MALDI-IHC.” In the Discussion, we described MALDI-IHC as an integration of MSI with immunohistochemistry using photocleavable mass-tagged antibodies. We have now cited a representative study: Yagnik et al. (2021, JASMS) which is the oldest citation we could find for the MALDI-IHC approach [12]. Additionally, we cite a Nature Medicine paper by Angelo et al. (2014) on multiplexed ion beam imaging (MIBI) [13] which uses metal-tagged antibodies and has parallels to MALDI-IHC. The reference has been added at the end of the sentence where MALDI-IHC is mentioned.

Revised text (Discussion): *“...MALDI-IHC, a recent innovation, integrates MALDI-MSI with immunohistochemistry by employing photocleavable mass-tagged antibodies [12]. A similar strategy was previously introduced as multiplexed ion beam imaging (MIBI) [13] which uses metal-tagged antibodies and has parallels to MALDI-IHC.”*

8) Figure S5 - Further explanation is warranted regarding the same ion image being shown with different potential identifications. Was anything done to try to confirm which (or more than one) image represents? Was/could immunohistochemistry be used for confirmation?

Thank you for pointing this out. We now have a new Supplementary Figure S6 and a new Table S3 obtained from the new imaging data acquired from the timsTOF flex that helped refine our list of candidates. The protein candidate(s) of interest can be validated by another technique such as immunohistochemistry.

Indeed we performed immunohistochemistry to validate two proteins. For proof of principle, we showed two examples, Nucleoprotein TPR and TBCA (Fig. 7 A-B). These two candidates were selected for validation based on supporting studies that show its relevance to thymic regeneration. We would like to respectfully point out that to validate all the proteins in the list may not be practical at this point and we accept this as a limitation of the paper.

Revised text (Results): *“In the updated timsTOF flex workflow, **Metric 4** is automated: the top 10 most abundant peptide ion images showing the largest differences between vehicle- (Ctrl) and Ctx-treated samples were selected without manual image review (Fig. S6). We retained a ± 1.0 Da mass tolerance to maximize sensitivity, though this occasionally resulted in duplicate assignments from isotopic peaks (e.g., both the monoisotopic and +1 Da isotope for Histone H4, Table S2). While more stringent processing such as deisotoping or narrower mass windows could reduce duplicates, these approaches risk false negatives given the current MALDI-MSI resolution. Isotopic matches were not removed but flagged in Table S2 to provide context for the user. After these filters were applied, the list of candidates was narrowed down to fewer candidates (Table S3 and Fig. S6). This to our knowledge, represents the largest spatiotemporally characterized proteome during endogenous thymic regeneration after chemotherapy insult. The protein candidate(s) of interest can be validated by another technique such as immunohistochemistry.”*

9) SI supplemental methods - The word "weather" should be "whether".

Apologies. This was very silly. It was corrected.

10) Figure S5 - More information is needed explaining why the same images are shown with multiple protein identifications. Which one or ones are correct and how have/can these be validated? In S5-16, the same protein is listed twice for the same peptide with different images, please confirm the listed m/z s and IDs

We have addressed this question in #8.

We hope the reviewers will find our edits to the manuscript satisfactory. All modifications in the manuscript are marked in the revised document for easy reference. We are grateful for the constructive feedback, which has undoubtedly strengthened our work. We hope that the revised manuscript is now suitable for publication in Life Science Alliance.

Sincerely,

George Karagiannis and Simone Sidoli

References:

1. Schober, Y., et al., *Protein identification by accurate mass matrix-assisted laser desorption/ionization imaging of tryptic peptides*. Rapid Commun Mass Spectrom, 2011. **25**(17): p. 2475-83.
2. Guo, G., et al., *Automated annotation and visualisation of high-resolution spatial proteomic mass spectrometry imaging data using HIT-MAP*. Nat Commun, 2021. **12**(1): p. 3241.
3. Kip, A.M., et al., *Combined Quantitative (Phospho)proteomics and Mass Spectrometry Imaging Reveal Temporal and Spatial Protein Changes in Human Intestinal Ischemia-Reperfusion*. J Proteome Res, 2022. **21**(1): p. 49-66.
4. Lemaire, R., et al., *Direct analysis and MALDI imaging of formalin-fixed, paraffin-embedded tissue sections*. J Proteome Res, 2007. **6**(4): p. 1295-305.
5. Fata, C.R., et al., *Are clear cell carcinomas of the ovary and endometrium phenotypically identical? A proteomic analysis*. Hum Pathol, 2015. **46**(10): p. 1427-36.
6. Dewez, F., et al., *MS Imaging-Guided Microproteomics for Spatial Omics on a Single Instrument*. Proteomics, 2020. **20**(23): p. e1900369.
7. Quanico, J., et al., *Combined MALDI Mass Spectrometry Imaging and Parafilm-Assisted Microdissection-Based LC-MS/MS Workflows in the Study of the Brain*. Methods Mol Biol, 2017. **1598**: p. 269-283.
8. Verenchikov, A.N., et al., *A Perspective of Multi-Reflecting TOF MS*. Mass Spectrom Rev, 2024.
9. Tsuji, Y., et al., *Mass Spectrometry Imaging (MSI) Delineates Thymus-Centric Metabolism In Vivo as an Effect of Systemic Administration of Dexamethasone*. Applied Sciences-Basel, 2021. **11**(22).
10. Denti, V., et al., *Lipidomic Typing of Colorectal Cancer Tissue Containing Tumour-Infiltrating Lymphocytes by MALDI Mass Spectrometry Imaging*. Metabolites, 2021. **11**(9).
11. Meyer, M.B., et al., *Spatial detection and consequences of nonrenal calcitriol production as assessed by targeted mass spectrometry imaging*. JCI Insight, 2024. **9**(15).
12. Yagnik, G., et al., *Highly Multiplexed Immunohistochemical MALDI-MS Imaging of Biomarkers in Tissues*. J Am Soc Mass Spectrom, 2021. **32**(4): p. 977-988.
13. Angelo, M., et al., *Multiplexed ion beam imaging of human breast tumors*. Nat Med, 2014. **20**(4): p. 436-42.

October 8, 2025

RE: Life Science Alliance Manuscript #LSA-2025-03205-TR

Dr. George Karagiannis
Albert Einstein College of Medicine
Microbiology and Immunology
1300 Morris Park Avenue
Forchheimer Building, Room 640
Bronx, NY 10461

Dear Dr. Karagiannis,

Thank you for submitting your revised manuscript entitled "Identifying Space-Resolved Proteins of the Murine Thymus, by Combining MALDI-MSI and Proteomics". Your manuscript was returned to Reviewers 2 and 3 and their reports are appended below.

As you will see Reviewer 2 is satisfied whereas Reviewer 3 made important suggestions to clarify methodology and improve the text. This reviewer also pointed out discrepancies in m/z and in peptide sequences related to Figure 3. We appreciate your confirmation to the journal that this was an error in manuscript preparation which will be resolved and the correct peptides, which already appeared in Supplementary Table 2 will be discussed in this section. We are extremely grateful to Reviewer 3 for noting this error. We would be happy to publish your paper in Life Science Alliance pending resolution of this point and the remaining requests by Reviewer 3, and final revisions necessary to meet our formatting guidelines.

- Please upload all figure files as individual ones, including the supplementary figure files; all figure legends should only appear in the main manuscript file.
- We encourage you to revise the figure legends for figures S1 and S5 such that the figure panels are introduced in alphabetical order.
- Please add a Summary Blurb/Alternate Abstract as well as the main Abstract in our system.
- Please add ORCID ID for secondary corresponding author--they should have received instructions on how to do so.
- The titles in both the system and the manuscript file must be consistent with each other.
- Please consult our manuscript preparation guidelines <https://www.life-science-alliance.org/manuscript-prep> and make sure your manuscript sections are in the correct order.
- LSA does not permit citation of "data not shown," "manuscript in preparation," "manuscript submitted," etc., in any section of the manuscript. Please either include data to support the claim on Line 627 or remove this claim.
- Please add an Author Contributions section to your main manuscript text
- Please add your main, supplementary figure, and table legends to the main manuscript text after the references section.
- Please add the X and Bluesky handles of your host institute/organization, as well as your own and/or one of the authors, in our system.
- Please add callouts for Figures 4C; S1B-F; S5A-C and S7 to your main manuscript text.
- Please move the move supplementary methods to the methods section in the main manuscript text.

A. FINAL FILES:

B. MANUSCRIPT ORGANIZATION AND FORMATTING:

Thank you for your attention to these final processing requirements. Please revise and format the manuscript and upload materials as soon as you are able.

Sincerely,

Reviewer #2 (Comments to the Authors (Required)):

The authors have appropriately addressed my initial review comments and I recommend publication. Below are minor comments the authors should consider.

- Line 56: "imaging mass spectrometry". Mass spectrometry imaging is used in the title and abstract. Further, it should be defined here first, not in the sentences below.
- Line 89: I would replace "profiled" with "investigated".

Reviewer #3 (Comments to the Authors (Required)):

In their resubmission of their manuscript, the authors have sought to address concerns addressed by reviewers in the previous version of the manuscript. They have added new experiments and made major changes, rendering it essentially a new submission. Unfortunately, these changes have not resulted in a manuscript that is suitable for publication. In fact, new and more serious problems have been introduced in this version of the manuscript, especially with regards to the peptides that the authors claim they have identified.

Major Concerns:

- 1) Very limited details are given in the methods section regarding sample preparation and data collection for MSI. What parameters were used on the sprayer? How long was the trypsin incubated? What were the mass spectrometer settings? These need to be included or at the very least referenced to another paper from which they were adapted. This is essential for reproduction of the experiment.
- 2) A mass tolerance of ± 1.0 Da is quite high, especially on the timsTOF fleX. When properly calibrated, the timsTOF should have a mass error of <10 ppm. This would correspond to 0.015 Da at m/z 1500. Much narrower tolerances should be used when working the timsTOF data.
- 3) The 3 main peptides that are discussed as being identified using the pepBridge approach (Figure 3) are not what the authors claim they are. They list m/z 946 as being sequence TKQTAR from Histone H2A. However, this sequence should have an m/z of 704.8 and the sequence is not found in Histone H2A. This sequence is found in Histone H3. Secondly, m/z 1337 is reported as sequence VVAFVANALAHKYH from Hemoglobin beta chain. This sequence is indeed found in hemoglobin beta (C-terminal peptide), but the m/z of this peptide should be 1436.80. Finally, they report m/z 1454 as being SDAAVDTSSEITK from Thymosin beta 4. However, this sequence is not found in thymosin beta 4 nor is the m/z of this sequence 1454. The m/z of this peptide should be 1424.67 and a blast search returned that this sequence is from prothymosin alpha. Based on typical MALDI-MSI experiments, it is unusual to detect lysine-terminated or C-terminal peptides as highly abundant and in FFPE tissue, lysine residues are likely to undergo formalin crosslinking. Arginine-terminated peptides are more basic and tend to ionize more readily than other peptides resulting in a sampling bias for arginine-terminated tryptic peptides. An additional concern is that none of these identified peptides are listed in the supplemental table of results from the combined MALDI-MSI and LC-MS/MS experiments, calling into question the validity of these results.

Minor Concerns:

- 1) Line 197 - The authors mention that the timsTOF has MALDI-2 and Trapped Ion Mobility, but it is unclear if either of these were used in the experiment.
- 2) Line 481 - The mass resolution of the timsTOF is $\sim 40,000$ while that of the MRT is $\sim 200,000$.
- 3) Line 632 - The authors continue to state "MALDI-MSI approaches have never before been applied to the thymus." This was shown to be false in the previously round of review of this manuscript.
- 4) Lines 675-676 - The timsTOF is not considered a "high resolution" instrument, especially when compared to an Orbitrap or an FT-ICR. It could at best be considered a medium resolution instrument.
- 5) Figure 6 - What peptide is shown in these MSI images? What m/z and sequence? This could be any peptide that shows approximately the same localization as KRT5.
- 6) Figure 7 - What peptides are shown in these MSI images? What m/z and sequence? Where are the insets taken from in the MSI images? This does nothing to convince the reader that the same things are being shown by MSI and IF.
- 7) Figure S5 - What peptide is shown in these MSI images? What sequence? Where are the insets taken from in the MSI images? This does nothing to convince the reader that the same things are being shown by MSI and IF.

Dear Dr. Fessenden,

Thank you for this new opportunity to complete our manuscript. We really appreciate the efforts of all reviewers, in particular Reviewer #3 who truly went out of his/her way to correct all our imprecisions. We are truly grateful for this level of attention, who helped us avoid some silly (although embarrassing) mistake.

We hope you will find this new version suitable for publication.

Simone Sidoli and George Karagiannis

Editor's comments

-LSA does not permit citation of "data not shown," "manuscript in preparation," "manuscript submitted," etc., in any section of the manuscript. Please either include data to support the claim on Line 627 or remove this claim.

Thank you for noting that. We removed the statement.

-Please add an Author Contributions section to your main manuscript text.

Sorry for forgetting. Here it is: *J.T.A. conceived and performed the experiments, analyzed the data, and prepared figures. C.M.-A. developed the pepBridge computational pipeline, conducted data integration, and contributed to data analysis. J.F. optimized MALDI-MSI acquisition parameters and assisted with spectral segmentation and image processing. G.S.K. and S.S. jointly supervised the project, provided conceptual guidance, and edited the manuscript. G.S.K. coordinated histopathology and immunofluorescence validation, while S.S. oversaw proteomics and mass spectrometry workflows. All authors discussed results and approved the final manuscript.*

-Please add your main, supplementary figure, and table legends to the main manuscript text after the references section.

Great! The captions are included.

-Please add callouts for Figures 4C; S1B-F; S5A-C and S7 to your main manuscript text.

Thank you for this note as well. We added them throughout the text.

-Please move the move supplementary methods to the methods section in the main manuscript text.

We moved the description of pepBridge from the supplementary material to the main manuscript.

Reviewer #2 (Comments to the Authors):

The authors have appropriately addressed my initial review comments and I recommend publication. Below are minor comments the authors should consider.

- Line 56: "imaging mass spectrometry". Mass spectrometry imaging is used in the title and abstract. Further, it should be defined here first, not in the sentences below.

We really appreciate the effort in reviewing our manuscript, and the kind note. We fixed this issue.

- Line 89: I would replace "profiled" with "investigated".

We fixed this as well. Thanks again!

Reviewer #3 (Comments to the Authors):

In their resubmission of their manuscript, the authors have sought to address concerns addressed by reviewers in the previous version of the manuscript. They have added new experiments and made major changes, rendering it essentially a new submission. Unfortunately, these changes have not resulted in a manuscript that is suitable for publication. In fact, new and more serious problems have been introduced in this version of the manuscript, especially with regards to the peptides that the authors claim they have identified.

Major Concerns:

1) Very limited details are given the methods section regarding sample preparation and data collection for MSI. What parameters were used on the sprayer? How long was the trypsin incubated? What were the mass spectrometer settings? These need to be included or at the very least referenced to another paper from which they were adapted. This is essential for reproduction of the experiment.

Thank you for this note. We modified the method section to add all these details. Please, see below the new parts colored in blue.

MALDI-MSI Analysis of Mouse Thymus Tissue Sections (Fig. 1A). *Thymi from vehicle- and cyclophosphamide (Ctx)-treated mice were fixed in 10% Neutral Buffered Formalin for 48h, processed, and paraffin embedded. FFPE thymic tissue sections, cut at 5- μ m thickness, were mounted on indium-tin oxide (ITO) conductive glass slides. The FFPE tissue sections were deparaffinized for 1.5 h at 60-65 °C and washed through a series of xylene, 100%, 95%, 70% ethanol and water. Tissue slides in 10 mM Tris buffer were heated to 95-98 °C using a steamer for 30 min. Slides were cooled down, buffer exchanged with water, and dried in a desiccator overnight. The tissue slides were then spray coated with trypsin solution using the TM Sprayer (HTX Technologies). Trypsin (0.05 μ g/ μ l; in 50 mM ammonium bicarbonate/10% acetonitrile) was sprayed on the tissue slide using the following parameters: nozzle temperature (30 oC), 10 passes, 0.025 mg/ml flow rate, 750 mm/min nozzle velocity, 2 mm track spacing, HH spray pattern, nitrogen gas pressure set at 10, nitrogen gas flow rate (3 L/min), 0 s drying time, 40 mm nozzle height. The total amount of trypsin was 20 μ g per slide. The slides were air dried for 5*

min and then incubated in a hydrated chamber for overnight trypsin digestion at 37 °C. After digestion, slides were dried for 15 min in the vacuum desiccator and then spray coated with CHCA matrix (α -cyano-hydroxycinnamic acid; 5 mg/ml in 50% AcN/0.2% TFA) using the TM Sprayer (HTX technologies) with the following settings: nozzle temperature (75 °C), 4 passes, 0.1 ml/min flow rate, 1300 mm/min nozzle velocity, 2 mm track spacing, CC spray pattern, nitrogen gas pressure set at 10, nitrogen gas flow rate (3 L/min), 10 s drying time, 40 mm nozzle height. The final matrix density was 0.00077 mg/mm² or 0.8 μ g/mm².

Peptide spatial mapping was performed by scanning and acquiring spectra m/z range of 700-3,500 at 100 μ m raster width using the Ultraflex extreme MALDI TOF/TOF mass spectrometer (Bruker Daltonics). Instrument parameters were set as follows: partial random walk, shots at raster spot: 49, diameter limit: 2,000 μ m, reflector gain: 2414 V, smart beam parameter: small, frequency: 1000 Hz, sample rate and digitizer: 1.00 Gs/s, ion source 1: 24.97 kV, ion source 2: 22.32 kV, lens: 7.41 kV, reflector: 26.54; reflector 2: 13.41 kV, matrix suppression: deflection, processing: SC peptide cent.

Spatial Proteomics. Peptide spatial mapping was performed by scanning and acquiring spectra m/z range of 700-3,500 at 50 μ m raster width timsTOF flex mass spectrometer (Bruker Daltonics) equipped with a MALDI-2 laser and Trapped Ion Mobility (TIM). Instrument parameters were set as follows (For these experiments, the Trapped Ion Mobility (TIMS) mode was active, while the MALDI-2 post-ionization source was not engaged): General: MALDI Plate Offset: 50 V, Deflection 1 Delta: 70 V, Funnel 1 RF: 500 Vpp, isCID Energy: 0 V, Funnel 2 RF: 350 V, Multipole RF: 400V, Collision Energy: 20 eV, Collision RF: 2500 Vpp, Quadrupole Ion Energy: 10 eV, Low Mass: 300 m/z , Transfer Time: 110 μ s, Pre Pulse Storage: 10 μ s. TIMS: deltaT1: -20 V, deltaT2: -120 V, deltaT3: 70 V, deltaT4: 100 V, deltaT5: 0V, deltaT6: 200V, Collision Cell In 220 V.

2) A mass tolerance of $\{\pm\}$ 1.0 Da is quite high, especially on the timsTOF flex. When properly calibrated, the timsTOF should have a mass error of <10 ppm. This would correspond to 0.015 Da at m/z 1500. Much narrower tolerances should be used when working the timsTOF data.

That is correct, apologies for being unclear. When using the timsTOF flex instrument, the actual mass tolerance used was set to 0.015 Da. However, for the metrics analysis, the mass window was set at the mass tolerance based on the Ultraflex extreme mass tolerance at 1 Da to show smaller delta mass can be obtained when a higher resolution timsTOF flex mass spectrometer is used compared to the Ultraflex extreme. In other words, we still wanted to display the same mass tolerance we used in the first manuscript submission to prove that the timsTOF flex has all the matches with a very narrow mass tolerance. In fact, in the second set of graphs, there is basically no match with a mass error LC-MS vs MALDI higher than 0.3 Da.

Please, see below Figure 3 from the first submission (based on the Metrics 1, 2, 3 from the Ultraflex extreme):

Below, we represent instead Figure 3 from the second submission (based on the Metrics 1,2,3 from the timsTOF flex data):

We hope this clarifies the doubt. Of course, we do not want to claim that the timsTOF flex has mass errors that reach 1 Da. We just wanted to display our metrics with the older instrumentation and show that with the newer equipment high mass errors lead to scores approaching zero.

3) The 3 main peptides that are discussed as being identified using the pepBridge approach (Figure 3) are not what the authors claim they are. They list m/z 946 as being sequence TKQTAR from Histone H2A. However, this sequence should have an m/z of 704.8 and the sequence is not found in Histone H2A. This sequence is found in Histone H3. Secondly, m/z 1337 is reported as sequence VVAFVANALAHKYH from Hemoglobin beta chain. This sequence is indeed found in hemoglobin beta (C-terminal peptide), but the m/z of this peptide should be 1436.80. Finally, they report m/z 1454 as being SDAAVDTSSEITK from Thymosin beta 4. However, this sequence is not found in thymosin beta 4 nor is the m/z of this sequence 1454. The m/z of this peptide should be 1424.67 and a blast search returned that this sequence is from prothymosin alpha. Based on typical MALDI-MSI experiments, it is unusual to detect lysine-terminated or C-terminal peptides as highly abundant and in FFPE tissue, lysine residues are likely to undergo formalin crosslinking. Arginine-terminated peptides are more basic and tend to ionize more readily than other peptides resulting in a sampling bias for arginine-terminated tryptic peptides. An additional concern is that none of these identified peptides are listed in the supplemental table of results from the combined MALDI-MSI and LC-MS/MS experiments, calling into question the validity of these results.

We are very sorry for this terrible mistake, and we are truly grateful for having spotted it. The reviewer saved ourselves from looking very silly in this circumstance. As we corresponded with the Editor, that paragraph went through a number of revisions, and somehow the final version contained a mistake. We are very sorry.

It was a very silly mistake because we know well the identifications of the signals in Figure 3; they are present in the manuscript (Supplementary Table 2). While we were addressing one of the reviewers' concerns, i.e. provide the identification of the signals extracted in Figure 3, we mixed up those masses with the wrong IDs. This is how that paragraph should read (fixed in the main text):

Our next step was to identify the major discriminating signals uncovered by these analyses. Unsupervised hierarchical clustering of the MALDI-MSI data yielded three major spectral

classes, characterized by m/z 946, 1337, and 1454 as distinguishing signals for the thymic medulla, cortex, and capsule regions, respectively (**Fig. 3A–D**). To link these cluster-specific ions to actual proteins, we cross-referenced the m/z values with our LC-MS/MS identifications. Notably, each could be assigned to a peptide: m/z 946 matched the $[M+H]^+$ of the Nucleoprotein TPR (sequence ESLLAEQR, monoisotopic 945.5 Da), m/z 1337 corresponded to a peptide from the Ras-related protein Rab-14 (sequence GAAGALMVYDITR, 1337.7 Da), and m/z 1454 aligned with Serotransferrin (sequence SKDFQLFSSPLGK, 1453.8 Da). These tentative identifications are reported in **Table S2**.

Minor Concerns:

1) Line 197 - The authors mention that the timsTOF has MALDI-2 and Trapped Ion Mobility, but it is unclear if either of these were used in the experiment.

Thank you for this note. That allowed us to be more specific in the method section. Please, see below:

“Spatial Proteomics. Peptide spatial mapping was performed by scanning and acquiring spectra m/z range of 700–3,500 at 50 μm raster width timsTOF flex mass spectrometer (Bruker Daltonics) equipped with a MALDI-2 laser and Trapped Ion Mobility (TIM). Instrument parameters were set as follows (For these experiments, the Trapped Ion Mobility (TIMS) mode was active, while the MALDI-2 post-ionization source was not engaged): General: MALDI Plate Offset: 50 V, Deflection 1 Delta: 70 V, Funnel 1 RF: 500 Vpp, isCID Energy: 0 V, Funnel 2 RF: 350 V, Multipole RF: 400V, Collision Energy: 20 eV, Collision RF: 2500 Vpp, Quadrupole Ion Energy: 10 eV, Low Mass: 300 m/z , Transfer Time: 110 μs , Pre Pulse Storage: 10 μs . TIMS: deltaT1: -20 V, deltaT2: -120 V, deltaT3: 70 V, deltaT4: 100 V, deltaT5: 0V, deltaT6: 200V, Collision Cell In 220 V.”

2) Line 481 - The mass resolution of the timsTOF is ~40,000 while that of the MRT is ~200,000.

Thank you. Indeed, we modified this statement as discussed in the next question #4.

3) Line 632 - The authors continue to state "MALDI-MSI approaches have never before been applied to the thymus." This was shown to be false in the previously round of review of this manuscript.

Apologies for missing this last statement. We removed it.

4) Lines 675-676 - The timsTOF is not considered a "high resolution" instrument, especially when compared to an Orbitrap or an FT-ICR. It could at best be considered a medium resolution instrument.

That is a fair point. We revised the statement with “...producing broader, less precise m/z peaks than newer systems such as multi-reflectron TOF instruments (Verenchikov et al., 2024) or the timsTOF flex (~60,000–80,000 resolution at m/z 400), which provides medium resolving power compared to Orbitrap or FT-ICR platforms.”

5) Figure 6 - What peptide is shown in these MSI images? What m/z and sequence? This could be any peptide that shows approximately the same localization as KRT5.

Apologies. We specified it better. For Krt 8, the peptide sequence is: AEAETMoxYQIK with oxidation on M (1199.39 m/z). For Krt5, the peptide sequence is: FVSTTSSSR (971.79 m/z).

We have added these in the figure captions on page 26 of the manuscript:

Figure 6. Correlation between MALDI-MSI and KRT5/8 Immunofluorescence. (A-B) KRT8 (AEAETMoxYQIK, 1199.39 m/z) and KRT5 (FVSTTSSSR, 971.79 m/z) signal constructed from MALDI-MSI molecular images (left) along with their paired KRT5 and KRT-8 immunofluorescence (right) on the sequential thymic sections of a cyclophosphamide-treated mouse (A), or a vehicle-treated mouse (B). Co-registration reveals mild shifts in the morphology and topography of the tissue, although most compartments can be captured. The yellow-dotted segments shown in the Ctrl animal on the right have been detached during the preparation and the processing of the MALDI-MSI and LC-MS/MS pipelines, and as such do not correspond to any segment on the MALDI-MSI acquisition. Impressively, the remaining component can be co-registered perfectly with the MALDI-MSI acquisition. Small-dotted boxes corresponding to the letters C, D, E, F, G and H, demonstrate the same fields-of-view in the MALDI-MSI acquisitions and the corresponding multiplex-IF sequential slides. (C-H) Magnified inserts of the six fields-of-view indicated in the dotted boxes in A and B, assessed by multiplexed immunofluorescence, for DAPI (nuclear stain), Keratin-5 (KRT5) and Keratin-8 (KRT8). C, E, and G correspond to the three fields-of-view in the cyclophosphamide (Ctx)-treated animal. D, F, and H correspond to the three fields-of-view in the vehicle (Ctrl)-treated animal. Hyphenated versions of these images, i.e., C', D', E', F', G', and H' represent magnified inserts of the white-dotted boxes in each corresponding figure, revealing in high-resolution and single-channels (top, KRT5; bottom, KRT8) the cellular details from their parental images. White-dotted lines correspond to the corticomedullary boundaries, as confirmed by KRT5 signal distribution and intensity, with the medulla staining as KRT5^{high} while other compartments as KRT5^{low}.

6) Figure 7 - What peptides are shown in these MSI images? What m/z and sequence? Where are the insets taken from in the MSI images? This does nothing to convince the reader that the same things are being shown by MSI and IF.

For the Tpr, the peptide sequence is: ELLAEQR (946.03 m/z). For the Tbca, the peptide sequence is: LEAAYTDLQILESEK (1850.88 m/z). We have added these in the figure captions on page 26 of the manuscript (below).

Regarding the insets, we thank the reviewer for raising this important point. It is indeed challenging to align slides that are sequentially or distally sectioned by several microns, especially when the modalities (MALDI-MSI and immunofluorescence) differ substantially in resolution and contrast. We have adopted standard practices from digital pathology and whole slide imaging literature, modified for our hybrid imaging, by using fiducial marks and tissue-based architectural landmarks (e.g. vasculature, capsule edges, cortical/medullary interfaces) to guide co-registration. These analyses were conducted by G.S.K., who has knowledge in veterinary pathology, and especially thymus pathology. While the two techniques rely on distinct labeling and imaging principles, the goal here is not sub-micron perfection, but to demonstrate that corresponding regions (particularly zonular boundaries like cortex-medulla or cortex-capsule) align closely enough to support our interpretation. To clarify, we have added a concise 2-3 sentence description of the co-registration strategy to the revised manuscript, in the section of immunofluorescence.

Figure 7. Immunolocalization of Novel Biomarkers of Thymic Compartmentalization in Murine Thymi at Cyclophosphamide-Induced Rebound Hyperplasia. Validation of two novel markers identified through our new MALDI-MSI pipeline, nucleoprotein Tpr (**A-D**), tubulin-specific chaperone-A (TBCA) (**E-H**). For each immunolabelled protein, a series of sequential representations is reported in a similar manner. The top rows represent cyclophosphamide (Ctx)-treated mice, while the bottom rows (hyphenated versions) represent vehicle (Ctrl)-treated mice. The first column reveals the molecular image of the respective protein, as obtained by MALDI-MSI acquisition, in this case, TPR (ESLLAEQR, 946.03 m/z) (A-A') and TBCA (LEAAYTDLQQILESEK, 1850.88 m/z) (E-E'). The second column reveals the immunolocalization of each respective protein and condition, using multiplex-immunofluorescence in paraffin-embedded tissues, and fields-of-view are representative of at least N=3, in this case, TPR (B-B') and TBCA (F-F'). In all cases, DAPI was used as nuclear stain, to document the presence of tissue, and demarcate the corticomedullary junction (white-dotted lines). C, Cortex; M, Medulla; white-dotted lines indicate corticomedullary junctions. In each image a white line (206 pixels for B, B', and 137 pixels for F, F') crosses through the corticomedullary junctions and is used for line profiling in the next column. In the third column, the normalized signal intensity of the corresponding protein is plotted along the line profile, in this case, TPR (C-C') and TBCA (G-G'). The black dotted lines indicate the pixel which segments the line into its cortical (left) and medullary (right) side. Intensity values are normalized to the maximum intensity observed from the two line-profiles of the same immunolabelled protein. The line profiles are representations of at least N=20 lines measured in each condition. The fourth and fifth columns represent higher magnifications of the region around the line profiles of the second column. They reveal at cellular detail the precise immunolocalization of each respective protein and condition, using multiplex-immunofluorescence in paraffin-embedded tissues, and fields-of-view are representative of at least N=10, in this case, TPR (D-D'), and TBCA (H-H'). For each example, the fourth column (left) is a merged image including DAPI as a nuclear stain, while the fifth column (right) is a single channel of the corresponding immunolabelled protein. White-dotted lines indicate the corticomedullary junction.

7) Figure S5 - What peptide is shown in these MSI images? What sequence? Where are the insets taken from in the MSI images? This does nothing to convince the reader that the same things are being shown by MSI and IF.

Sorry, we fixed this too. For Krt5, the peptide sequence is: FVSTTSSSR (971.79 m/z).

Supplementary Figure S5. MALDI KRT5 (FVSTTSSSR, 971.79 m/z) (vs immunohistochemistry comparison).

With regard to the insets, please refer to our response to the previous comment. Concerning Supplementary Figure 5, however, we would like to clarify that the immunofluorescence (IF) image was obtained from a different thymus sample than the one used for MALDI-MSI. This IF imaging serves mostly to confirm that in two independent thymus cases, one treated with vehicle (Fig S5B) and one with Ctx (Fig S5C), KRT8 and KRT5 expression have the expected localization in the thymic compartments, as the ones indicated from the co-registered cases, i.e. Fig. 6. We have also slightly modified the manuscript in the first paragraph of the section "Validation of MALDI-MSI KRT8 Mapping by Immunofluorescence"

October 21, 2025

RE: Life Science Alliance Manuscript #LSA-2025-03205-TRR

Dr. George Karagiannis
Albert Einstein College of Medicine
Microbiology and Immunology
1300 Morris Park Avenue
Forchheimer Building, Room 640
Bronx, NY 10461

Dear Dr. Karagiannis,

Thank you for submitting your Methods entitled "Identifying Space-Resolved Proteins of the Murine Thymus, by Combining MALDI-MSI and Proteomics". It is a pleasure to let you know that your manuscript is now accepted for publication in Life Science Alliance. Congratulations on this interesting work and thank you for addressing the remaining reviewer comments in particular those of Reviewer 3.

DISTRIBUTION OF MATERIALS:

Again, congratulations on a very nice paper. I hope you found the review process to be constructive and are pleased with how the manuscript was handled editorially. We look forward to future exciting submissions from your lab.

Sincerely,
